# PN-GAIL: Leveraging Non-optimal Information from Imperfect Demonstrations

**Qiang Liu, Huiqiao Fu, Kaiqiang Tang & Chunlin Chen**[*]
School of Management and Engineering
Nanjing University
Nanjing, China
{qiangliu, hqfu, kqtang}@smail.nju.edu.cn, clchen@nju.edu.cn

**Daoyi Dong**
The Australian Artificial Intelligence Institute
University of Technology Sydney
Sydney, Australia
daoyidong@gmail.com

## Abstract

Imitation learning aims at constructing an optimal policy by emulating expert demonstrations. However, the prevailing approaches in this domain typically presume that the demonstrations are optimal, an assumption that seldom holds true in the complexities of real-world applications. The data collected in practical scenarios often contains imperfections, encompassing both optimal and non-optimal examples. In this study, we propose Positive-Negative Generative Adversarial Imitation Learning (PN-GAIL), a novel approach that falls within the framework of Generative Adversarial Imitation Learning (GAIL). PN-GAIL innovatively leverages non-optimal information from imperfect demonstrations, allowing the discriminator to comprehensively assess the positive and negative risks associated with these demonstrations. Furthermore, it requires only a small subset of labeled confidence scores. Theoretical analysis indicates that PN-GAIL deviates from the non-optimal data while mimicking imperfect demonstrations. Experimental results demonstrate that PN-GAIL surpasses conventional baseline methods in dealing with imperfect demonstrations, thereby significantly augmenting the practical utility of imitation learning in real-world contexts. Our codes are available at https://github.com/QiangLiuT/PN-GAIL.

## 1 Introduction

In recent years, Reinforcement Learning (RL) has achieved significant success in addressing sequential decision-making problems (Sutton & Barto, 2018; Xia et al., 2020; Zha et al., 2021). Its primary goal is to optimize policies to maximize cumulative rewards. However, designing an appropriate reward function can be quite challenging; a poorly designed reward function can lead to suboptimal performance of RL agents. In contrast, Imitation Learning (IL) presents a more practical approach, as it learns solely from demonstrations, eliminating the need for explicitly defined reward functions. Generative Adversarial Imitation Learning (GAIL) (Ho & Ermon, 2016), which employs the framework of Generative Adversarial Networks (GANs) (Goodfellow et al., 2014), directly learns a policy from demonstrations. Following the development of GAIL, many variants have been proposed to enhance algorithmic performance across different problem domains (Li et al., 2017; Fu et al., 2018; Dadashi et al., 2020; Fu et al., 2024).

The imitation learning methods mentioned above can learn an optimal policy given optimal demonstrations. However, most imitation learning methods tend to fail when faced with data filled with imperfect demonstrations. Especially in the real world, the assumption that the provided demonstrations are of high quality may not always be valid (Yang et al., 2024). For instance, due to factors

---

[*]Corresponding author

such as fatigue and distractions, decisions made by human experts may not always be optimal. In such cases, simply assigning equal weight to all data can lead to a decrease in the quality of the learned policy. Therefore, we need a method that can extract useful information from imperfect demonstrations to learn an optimal policy.

Existing methods for imitation learning from imperfect demonstrations can be broadly divided into two categories: weighting-based methods (Wu et al., 2019; Wang et al., 2021b;a; Tangkaratt et al., 2020; Zhang et al., 2021; Wang et al., 2023) and ranking-based methods (Brown et al., 2019; 2020; Chen et al., 2021; Huo et al., 2023; Taranovic et al., 2022). Weighting-based methods achieve imitation of optimal demonstrations through reweighting different demonstrations, while ranking-based methods aim to guide the recovery of the reward function with additional ranking information, thereby learning an optimal policy based on the rewards. In contrast, weighting-based methods are more computationally efficient since they do not require trajectory sorting. Additionally, they are more flexible to use, as they do not necessitate demonstrations to be in a trajectory form.

In order to solve the problem of learning from imperfect demonstrations using GAIL, Wu et al. (2019) proposed two methods: two-step importance weighting IL (2IWIL) and generative adversarial IL with imperfect demonstration and confidence (IC-GAIL). The former trains a classifier to forecast confidence scores and subsequently proceeds with weighted imitation learning, employing a two-step learning approach. The latter introduces an end-to-end learning method but at a slower pace of learning. However, as discussed in Section 4.1, 2IWIL is susceptible to the influence of preferences inherent in imperfect demonstrations during training. In the learning process of the discriminator, 2IWIL tends to assign a higher "reward" to the state-action pair with a greater probability of occurrence in imperfect demonstrations. This discrepancy in "rewards" diverges from our intended objectives, potentially resulting in the acquisition of a suboptimal policy.

To tackle the aforementioned challenge, we propose a new method, Positive-Negative Generative Adversarial Imitation Learning (PN-GAIL), building upon the framework of GAIL. Different from 2IWIL, we leverage non-optimal information from imperfect demonstrations, enabling the discriminator to weigh both positive and negative risks of imperfect demonstrations comprehensively and requiring only a small subset of labeled confidence scores. In this way, it can provide more accurate rewards for subsequent RL methods. Theoretical analysis reveals that PN-GAIL not only mimics imperfect demonstrations but also avoids imitating non-optimal ones, illustrating the ability of PN-GAIL to learn an optimal policy. Additionally, to get more accurate confidence scores, we propose an improved semi-supervised confidence classifier. Experiments on six control tasks are conducted to show the efficiency of our method in dealing with imperfect demonstrations compared to baseline methods. In particular, the main contributions of this work are threefold:

1. We propose a new method called PN-GAIL, which can leverage non-optimal information to learn an optimal policy from imperfect demonstrations.

2. We theoretically analyze the output of the optimal discriminator in PN-GAIL, demonstrating that PN-GAIL learns an optimal policy by deviating from the non-optimal demonstrations.

3. We demonstrate the efficiency of our method across six control tasks, with results showing superior performance compared to other baseline methods.

## 2 RELATED WORK

**Imitation Learning** Imitation learning methods can learn an optimal policy when given optimal demonstrations. Behavior cloning (BC) (Pomerleau, 1988) learns policies directly through a supervised learning paradigm and is mostly used in autonomous driving tasks (Hawke et al., 2020). While straightforward, it suffers from compounded errors due to covariate shift (Ross & Bagnell, 2010) and typically demands extensive data for effective training. Inverse Reinforcement Learning (IRL) (Abbeel & Ng, 2004; Ziebart et al., 2008) first seeks to recover the underlying reward function and then learns a policy through RL. On the other hand, GAIL views an imitation learning problem through the lens of occupancy measures (Puterman, 2014), and can learn a policy directly from the demonstrations. GAIL has demonstrated success across various imitation tasks, including multi-agent scenarios (Song et al., 2018), robot control (Peng et al., 2021), human motion simulation (Wei et al., 2021), and imitation of driver behavior (Bhattacharyya et al., 2022; Ruan & Di, 2022). How-

ever, these methods presuppose access to optimal demonstrations. When provided with imperfect demonstrations, they struggle to learn a good policy.

**Weighting-based imitation learning from imperfect demonstrations** Weighting-based imitation learning from imperfect demonstrations learns an optimal policy by reweighting different demonstrations and amplifying the significance of the optimal ones. 2IWIL and IC-GAIL (Wu et al., 2019) first propose to reweight imitation learning based on confidence. WGAIL (Wang et al., 2021b) connects confidence with the agent policy and discriminator without requiring additional prior information on confidence. However, it needs a high proportion of optimal demonstrations in imperfect demonstrations. VILD (Tangkaratt et al., 2020) employs a variational method to jointly estimate demonstration quality and reward, but it assumes that the quality of demonstrations be correlated with variance. CAIL (Zhang et al., 2021) guides confidence estimation by introducing trajectory ranking. UID (Wang et al., 2023) treats imperfect demonstrations as unlabeled data, based on the idea of PU Learning (Du Plessis et al., 2014), mitigating the impact of non-optimal demonstrations. Nevertheless, this relies on the assumption that non-optimal demonstrations within the imperfect demonstrations can well match agent demonstrations. Additionally, some studies address imperfect demonstrations in offline imitation learning (Sasaki & Yamashina, 2020; Xu et al., 2022; Kim et al., 2021; Yu et al., 2023; Li et al., 2024). However, these methods either similarly assume that the proportion of the optimal demonstrations is dominant, or require an additional set of optimal demonstrations.

**Ranking-based imitation learning from imperfect demonstrations** Ranking-based imitation learning from imperfect demonstrations utilizes additional ranking information to guide the recovery of the reward function, thereby learning a policy based on the rewards. T-REX (Brown et al., 2019) infers the reward function from the given ranking trajectories and expects the reward function to conform to the given ranking order. However, this approach demands a substantial quantity of ranking trajectories to enhance its generalization capacity. D-REX (Brown et al., 2020) automatically generates ranking trajectories by introducing varying degrees of noise. SSRR (Chen et al., 2021) revises the structure of the reward function in D-REX to accommodate different levels of noise influence better. LERP (Huo et al., 2023) views suboptimal demonstrations as additive noise on the reward function, establishing a quantifiable relationship between noise and reward based on D-REX. However, the automatic generation of the ranking trajectories requires the assumption that the trajectory will receive lower rewards with the addition of noise, which is not necessarily true in cases where random demonstrations exist. AILP (Taranovic et al., 2022) necessitates the teacher's access to the true reward function, thereby providing real-time correct ranking between two trajectories. Nevertheless, this condition is challenging to meet in practice.

## 3 PRELIMINARIES

In this section, we provide a brief background on RL, GAIL, and 2IWIL.

**Reinforcement learning** We consider the standard Markov Decision Process (MDP) (Sutton & Barto, 2018). An MDP typically comprises six components, denoted as $\mathcal{M} = \langle \mathcal{S}, \mathcal{A}, \mathcal{P}, \mathcal{R}, \rho_0, \gamma \rangle$, where $\mathcal{S}$ is the state space, $\mathcal{A}$ is the action space, $\mathcal{P}(s_{t+1}|s_t, a_t)$ is the transition probability from state $s_t$ and action $a_t$ at time step $t$ to state $s_{t+1}$ at time step $t+1$, $\mathcal{R}(s, a)$ is the reward function, $\rho_0$ is the distribution of initial states, and $\gamma \in (0, 1)$ stands for the discount factor. In an RL process, the agent aims to learn a policy $\pi(a|s)$ to maximize its expected discounted rewards $\mathbb{E}_{s_0 \sim \rho_0, \pi} \left[ \sum_{t=0}^{\infty} \gamma^t \mathcal{R}(s_t, a_t) \right]$. For any given policy $\pi$, there exists a corresponding occupancy measure $\rho_\pi : S \times \mathcal{A} \to \mathbb{R}$, establishing a one-to-one relationship between them.

**GAIL and 2IWIL** GAIL integrates GANs framework into imitation learning, leading the following min-max optimization problem by minimizing the Jensen-Shannon divergence between $p_\theta$ and $p_E$ (Ke et al., 2021):

$$\min_\theta \max_w \mathbb{E}_{(s,a) \sim p_\theta}[\log D_w(s, a)] + \mathbb{E}_{(s,a) \sim p_E}[\log(1 - D_w(s, a))], \qquad (1)$$

where $p_\theta$ and $p_E$ are the corresponding normalized occupancy measures for the agent policy $\pi_\theta$ and the expert policy $\pi_E$, respectively. The discriminator $D_w$ attempts to discern these distributions from

$\pi_{\mathrm{E}}$ and $\pi_\theta$, while $\pi_\theta$ aims to "trick" the discriminator, thereby minimizing $\mathbb{E}_{(s,a)\sim p_\theta}[\log D_w(s,a)]$. Ultimately, the output of the discriminator, $-\log D_w(s,a)$, serves as a reward, which can then be utilized to learn the policy $\pi_\theta$ through RL methods such as TRPO (Schulman et al., 2015), PPO (Schulman et al., 2017) and SAC (Haarnoja et al., 2018).

Since GAIL assigns the same weights to all demonstrations, if the given demonstrations are non-optimal, then the learned policy will also be non-optimal. To address this issue, 2IWIL considers the following setup:

$$\mathcal{D}_{\mathrm{c}} \triangleq \{(x_{\mathrm{c},i}, r_i)\}_{i=1}^{n_{\mathrm{c}}} \overset{\mathrm{i.i.d.}}{\sim} q(x,r),$$

$$\mathcal{D}_{\mathrm{u}} \triangleq \{x_{\mathrm{u},i}\}_{i=1}^{n_{\mathrm{u}}} \overset{\mathrm{i.i.d.}}{\sim} p(x),$$

where $x$ is the state-action pair, $r$ denotes confidence score, indicating the probability that $x$ belongs to the optimal demonstrations, $q(x,r) = p(x)p_{\mathrm{r}}(r|x)$ and $p_{\mathrm{r}}(r_i|x) = \delta(r_i - r(x))$ is Dirac delta function. $\mathcal{D}_{\mathrm{c}}$ and $\mathcal{D}_{\mathrm{u}}$ represent confidence data and unlabeled data, respectively.

2IWIL first trains a probabilistic classifier, which forecasts the confidence scores of demonstrations in $\mathcal{D}_{\mathrm{u}}$ through *semi-conf (SC) classification*, leveraging the knowledge of confidence scores in $\mathcal{D}_{\mathrm{c}}$. The probabilistic classifier is trained with the loss function as follows:

$$R_{\mathrm{SC},\ell}(g) = \mathbb{E}_{x,r\sim q}\left[r\ell(g(x)) + (1-r)\ell(-g(x)) - \beta\ell(-g(x))\right] + \mathbb{E}_{x\sim p}[\beta\ell(-g(x))], \quad (2)$$

where $g$ is a prediction function, $\ell$ is a loss function which uses logistic loss and $\beta = \frac{n_{\mathrm{u}}}{n_{\mathrm{c}}+n_{\mathrm{u}}}$. After obtaining confidence scores for all demonstrations, 2IWIL uses Bayes' rule to reweight the GAIL objective. The final objective becomes

$$\min_\theta \max_w \mathbb{E}_{x\sim p_\theta}\left[\log D_w(x)\right] + \mathbb{E}_{x\sim p}\left[\frac{r(x)}{\eta}\log(1 - D_w(x))\right], \quad (3)$$

where $\eta$ is a class-prior, denoting the proportion of optimal demonstrations within the imperfect demonstrations, and $p$ is the corresponding normalized occupancy measures for $\mathcal{D}_{\mathrm{c}}$ and $\mathcal{D}_{\mathrm{u}}$.

## 4 APPROACH

In this section, we begin by elucidating the motivation behind our method. We illustrate the problem of 2IWIL through an example and then introduce our method PN-GAIL with theoretical analysis. Details of derivations and proofs in this section can be found in Appendix B.

### 4.1 MOTIVATION

2IWIL aims to reweight demonstrations based on confidence, assigning greater weights to those with high confidence so that the discriminator can give higher rewards. However, it is worth noting that this weighting behavior can be influenced by the preferences inherent in imperfect demonstrations. As shown in Fig. 1, the top half of the graph represents the actual confidence scores, while the bottom half represents the equivalent weights during the discriminator's training. When imperfect demonstrations favor a low-confidence state-action pair, we consider the goals of GAIL, 2IWIL:

$$\min_\theta \max_w \mathbb{E}_{x\sim p_\theta}\left[\log D_w(x)\right] + \mathbb{E}_{x\sim p}\left[\frac{r(x)}{\eta}\log(1 - D_w(x))\right].$$

GAIL assigns the same weights to all given demonstrations, which means $\frac{r(x)}{\eta} \equiv 1.0$. For the given expert demonstrations, we only need to consider the second term of the above equation, which can be expanded as: $\sum p(x)\frac{r(x)}{\eta}\log(1 - D_w(x))$. Here, the coefficient of $\log(1 - D_w(x))$ is $\frac{r(x)}{\eta}$. Therefore, if there is a higher probability of $x_1$ appearing in imperfect demonstrations, e.g., $p(x_1) = 5p(x_{other})$ (assuming that the probabilities of other state-action pairs are the same), then, for $x_1$, the coefficient of $\log(1 - D_w(x))$ is $p(x_1)\frac{r(x_1)}{\eta} = p(x_{other})\frac{5r(x_1)}{\eta}$. This can also be explained as that the probability of $x_1$ appearing is the same as for other demonstrations, but the confidence score is 5 times higher, since $\eta$ is constant across all demonstrations. In the case of GAIL, since $\frac{r(x)}{\eta} \equiv 1.0$, the confidence score becomes 5 times of the original, which is calculated as $1.0 \times 5 = 5.0$. This

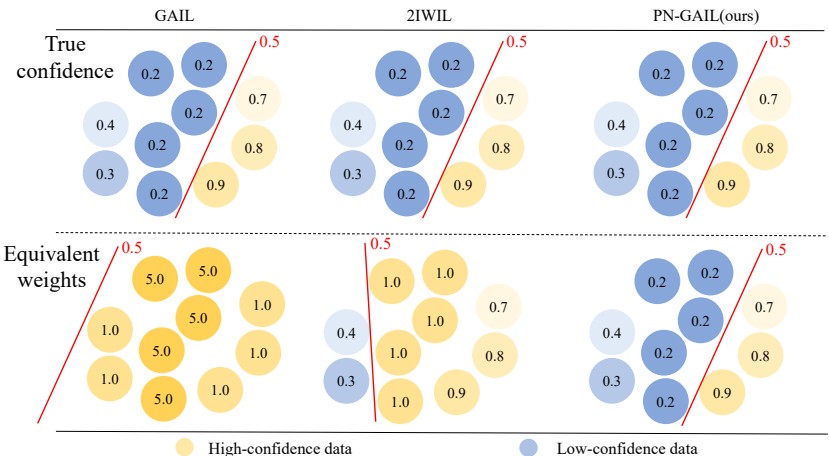

Figure 1: Schematic diagram of the difference between PN-GAIL, 2IWIL and GAIL. The top half of the graph is the actual confidence score, the bottom half is the equivalent weight when training the discriminator, and the red line is distinguished by a threshold of $0.5$.

can lead to low-confidence data being treated as high-confidence data, not aligning with the actual situation. In addition, for a clearer explanation, we also provide a simple example.

Suppose a state $s_1$ has two actions $x_1(s_1, a_1)$ and $x_2(s_1, a_2)$. In Fig. 1, the circle with a confidence score of $0.8$ represents $x_2$, and the five circles with a confidence score of $0.2$ all represent $x_1$, which means $p(x_1) = 5p(x_2)$, indicating a higher probability of $x_1$ occurring in imperfect demonstrations. It is clear that $x_2$ is better than $x_1$. However, in imperfect demonstrations where $p(x_1) = 5p(x_2)$ and the prior $\eta$ is the same, according to Eq. (3), the equivalent weight of $x_1$ will be $1.0$ compared to $x_2$ ($0.2 \times 5 = 1.0$). This means that the discriminator will consider $x_1$ to be more likely the optimal demonstration than $x_2$, resulting in a poor policy.

### 4.2 POSITIVE-NEGATIVE GENERATIVE ADVERSARIAL IMITATION LEARNING

To tackle the problem above, we propose Positive-Negative Generative Adversarial Imitation Learning (PN-GAIL). This method leverages non-optimal information from imperfect demonstrations, allowing the discriminator to comprehensively assess the positive and negative risks associated with these demonstration. By doing so, it mitigates the influence of preferences inherent in imperfect demonstrations on the discriminator, thus ensuring that its evaluations better reflect actual conditions. This, in turn, provides more accurate rewards for the subsequent RL process, leading to the learning of a better policy.

We begin by focusing on the training of the discriminator, denoting optimal demonstrations as positive examples and non-optimal demonstrations as negative examples. In 2IWIL, the discriminator only considers the positive risk of imperfect demonstrations, while ignoring negative risk. Therefore, the discriminator will heavily prioritize the positive risk training for state-action pairs frequently appearing in imperfect demonstrations, leading to incorrect results. For this reason, we aim to incorporate the negative risk into the training of the discriminator when dealing with imperfect demonstrations. Specifically, following Xu & Denil (2021), let $(X, Y)$ represent the input and output of a binary classification problem, where $X$ denotes the state-action pair and $Y \in \{0, 1\}$. We label optimal data as $0$ and non-optimal data as $1$. The imperfect demonstrations is denoted as $\mathcal{D}$, comprising $\mathcal{D}_{\text{opt}}$(optimal demonstrations) and $\mathcal{D}_{\text{non}}$(non-optimal demonstrations), where $\mathcal{D} = \mathcal{D}_{\text{opt}} + \mathcal{D}_{\text{non}}$. We aim to train a discriminator $D_w$ using a loss function $\phi : \mathbb{R} \times \{0, 1\} \to \mathbb{R}$. Utilizing the labeled risk operator as follows:

$$R^y_{D_w}(\mathcal{D}) = \mathbb{E}_{\mathcal{D}}[\phi(D_w(x), y)]. \tag{4}$$

We expect the discriminator to provide accurate evaluation scores for both the dataset generated by the agent policy and the given imperfect demonstrations. To achieve this, we consider the risk associated with the dataset generated by the agent policy and the risk associated with the imperfect

demonstrations, respectively. The overall risk of the discriminator is

$$R_{D_w}^{pn}(\mathcal{D}_{\pi_\theta}, \mathcal{D}) = R_{D_w}^1(\mathcal{D}_{\pi_\theta}) + R_{D_w}^{pn}(\mathcal{D}), \tag{5}$$

where $\mathcal{D}_{\pi_\theta}$ is the demonstrations generated by agent policy $\pi_\theta$. We can write the risk associated with the imperfect demonstrations as the sum of positive and negative risks:

$$R_{D_w}^{pn}(\mathcal{D}) = R_{D_w}^{pn}(\mathcal{D}_{\text{opt}}, \mathcal{D}_{\text{non}}) = \eta R_{D_w}^0(\mathcal{D}_{\text{opt}}) + (1 - \eta) R_{D_w}^1(\mathcal{D}_{\text{non}}), \tag{6}$$

where $\eta = p(y = 0)$ is a class-prior, denoting the proportion of optimal demonstrations within the imperfect demonstrations.

Based on Eq. (6), the overall risk of the discriminator can be rewritten as

$$R_{D_w}^{pn}(\mathcal{D}, \mathcal{D}_{\pi_\theta}) = R_{D_w}^1(\mathcal{D}_{\pi_\theta}) + \eta R_{D_w}^0(\mathcal{D}_{\text{opt}}) + (1 - \eta) R_{D_w}^1(\mathcal{D}_{\text{non}}). \tag{7}$$

Replacing the loss function with the standard logistic loss and tidying up the statement, the objective of the discriminator becomes

$$\max_w \mathbb{E}_{x \sim p_\theta}[\log D_w(x)] + \eta \mathbb{E}_{x \sim p_{\text{opt}}}[\log(1 - D_w(x))] + (1 - \eta)\mathbb{E}_{x \sim p_{\text{non}}}[\log D_w(x)]. \tag{8}$$

Since $r(x)$ denotes the probability that $x$ belongs to the optimal demonstrations, which means $r(x) = p(y = 0|x)$ and $1 - r(x) = p(y = 1|x)$, according to the Bayes' rule we have

$$p_{\text{opt}}(x) = p(x|y = 0) = \frac{r(x)p(x)}{\eta}, \quad p_{\text{non}}(x) = p(x|y = 1) = \frac{(1 - r(x))p(x)}{1 - \eta}. \tag{9}$$

Then we can rewrite the objective of the discriminator in the following theorem.

**Theorem 4.1.** *Based on Eq. (9), the objective of the discriminator can be rewritten as*

$$\max_w \mathbb{E}_{x \sim p_\theta}[\log D_w(x)] + \mathbb{E}_{x \sim p}[r(x)\log(1 - D_w(x))] + \mathbb{E}_{x \sim p}[(1 - r(x))\log D_w(x)]. \tag{10}$$

The agent receives a reward equivalent to $-\log D_w(x)$, and then the final objective to be optimized becomes

$$\min_\theta \max_w \mathbb{E}_{x \sim p_\theta}[\log D_w(x)] + \mathbb{E}_{x \sim p}[r(x)\log(1 - D_w(x))] + \mathbb{E}_{x \sim p}[(1 - r(x))\log D_w(x)]. \tag{11}$$

Furthermore, recall that 2IWIL adopts a two-step learning approach, where $\mathcal{D}_c$ and $\mathcal{D}_u$ represent confidence data and unlabeled data, respectively. To get more accurate confidence scores, we refine the *semi-conf (SC) classification* proposed in 2IWIL, which is trained by minimizing the following risk:

$$R_{\text{SC},\ell}(g) = \mathbb{E}_{x,r \sim q}[r\ell(g(x)) + (1 - r)\ell(-g(x)) - \beta\ell(-g(x))] + \mathbb{E}_{x \sim p}[\beta\ell(-g(x))]. \tag{12}$$

We note that for a state-action pair $x$ occurring solely in $\mathcal{D}_c$, once $1 - r - \beta < 0$, where $r > 1 - \beta$, the coefficient of $\ell(-g(x))$ becomes negative. In order to minimize the risk, the classifier would then forecast $g(x)$ as positive infinity, leading to an excessively high estimation of confidence for demonstrations in $\mathcal{D}_c$. Concurrently, Eq. (12) tends to predict data in $\mathcal{D}_u$ as negative, resulting in an underestimated confidence for demonstrations in $\mathcal{D}_u$. To balance this effect, we propose *balanced semi-conf (BSC) classification*. We introduce $\mathbb{E}_{x \sim p}[\alpha\ell(g(x))] - \mathbb{E}_{x \sim q}[\alpha\ell(g(x))]$, the theoretical value of which is 0 since $\mathcal{D}_c$ and $\mathcal{D}_u$ are drawn from the same distribution $p(x)$. The final risk is as follows:

$$R_{\text{BSC},\ell}(g) = \mathbb{E}_{x,r \sim q}[r\ell(g(x)) + (1 - r)\ell(-g(x)) - \alpha\ell(g(x)) - \beta\ell(-g(x))]$$
$$+ \mathbb{E}_{x \sim p}[\alpha\ell(g(x)) + \beta\ell(-g(x))], \tag{13}$$

where the loss function $\ell$ uses the logistic loss. Next, similar to 2IWIL, we seek to derive the optimal values of $\alpha$ and $\beta$ for minimizing the variance of the empirical unbiased estimator $\widehat{R}_{\text{BSC},\ell}(g)$ through the following theorem.

**Theorem 4.2.** *Let $d_1$ denote $\text{Var}(\ell(-g(x)))$, $d_2$ denote $\text{Var}(\ell(g(x)))$, $\sigma_{\text{cov1}}$ denote the covariance between $\frac{1}{n_c}\sum_{i=1}^{n_c} r_i(\ell(g(x_{c,i})) - \ell(-g(x_{c,i})))$ and $\frac{1}{n_c}\sum_{i=1}^{n_c}\ell(-g(x_{c,i}))$, $\sigma_{\text{cov2}}$ denote the covariance between $\frac{1}{n_c}\sum_{i=1}^{n_c}(1 - r_i)(\ell(-g(x_{c,i})) - \ell(g(x_{c,i})))$ and $\frac{1}{n_c}\sum_{i=1}^{n_c}\ell(g(x_{c,i}))$, cov denote $\text{Cov}(\ell(-g(x)), \ell(g(x)))$. The estimator $\widehat{R}_{\text{BSC},\ell}(g)$ has the minimum variance when*

$$\alpha = \frac{n_u}{n_c + n_u} - \frac{d_1 cov - cov^2}{d_1 d_2 - cov^2}\frac{n_u}{n_c + n_u} + \frac{d_1\sigma_{\text{cov2}} - cov\sigma_{\text{cov1}}}{d_1 d_2 - cov^2}\frac{n_c n_u}{n_c + n_u},$$

$$\beta = \frac{n_u}{n_c + n_u} - \frac{d_2 cov - cov^2}{d_1 d_2 - cov^2}\frac{n_u}{n_c + n_u} + \frac{d_2\sigma_{\text{cov1}} - cov\sigma_{\text{cov2}}}{d_1 d_2 - cov^2}\frac{n_c n_u}{n_c + n_u}.$$

Since $d_1, d_2, \sigma_{\text{cov1}}, \sigma_{\text{cov2}}, cov$ are difficult to calculate, in practice, we assume that these covariances are sufficiently small for computational convenience. Consequently, we have $\alpha = \frac{n_u}{n_c + n_u}$ and $\beta = \frac{n_u}{n_c + n_u}$. During the training process, as we assume that the data from $\mathcal{D}_c$ and $\mathcal{D}_u$ are drawn from the same distribution $p(x)$, we guarantee this condition via the clip function (see more details in Appendix C.2).

## 4.3 THEORETICAL ANALYSIS

We consider the reward given by the optimal discriminator $D_w^*(x)$. In 2IWIL, when the discriminator is optimal, the reward is $-\log D_w^*(x) = \log\left((rp + \eta p_\theta)/(\eta p_\theta)\right)$. Consequently, if imperfect demonstrations exhibit a pronounced preference for a certain state-action pair, it results in a significantly higher probability of $p$ compared to other state-action pairs. The discriminator tends to provide an inflated reward, hindering the learning of an optimal policy. Conversely, in our method, we first give the following theorem:

**Theorem 4.3.** *Given a fixed agent policy $\pi_\theta$, the optimal discriminator $D_w^*(x)$ of Eq. (11) can be written as*

$$D_w^*(x) = \frac{(1 - r)p + p_\theta}{p + p_\theta}. \tag{14}$$

*As a result, when the optimal discriminator $D_w^*(x)$ is given, the optimization of $\pi_\theta$ is equivalent to minimizing*

$$2\text{JSD}(p_\theta || p) - \text{KL}(p_\theta || p_1) - (1 - \eta)\text{KL}(p_{\text{non}} || p_1) + C, \tag{15}$$

*where $p_1 = (p_\theta + (1 - \eta)p_{\text{non}})/(2 - \eta)$, $C = \eta\mathbb{E}_{x \sim p_{\text{opt}}}\left[\log \frac{\eta p_{\text{opt}}}{p}\right] + (1 - \eta)\mathbb{E}_{x \sim p_{\text{non}}}\left[\log \frac{(1-\eta)p_{\text{non}}}{p}\right] + \log(2 - \eta) - (1 - \eta)\log(1 - \eta)/(2 - \eta) - 2\log 2$, which is a constant for $\pi_\theta$.*

According to Theorem 4.3, since $p_1$ is a weighted sum of $p_\theta$ and $p_{\text{non}}$, subtracting the second and third terms of the Kullback-Leibler (KL) divergence is equivalent to letting $p_\theta$ deviate from $p_{\text{non}}$. Thus, PN-GAIL aims to align $p_\theta$ with $p$ and ensure that $p_\theta$ deviates from $p_{\text{non}}$. This illustrates that our method is able to avoid mimicking non-optimal data within imperfect demonstrations, thereby solely imitating the optimal ones. Additionally, the reward given by the optimal discriminator $D_w^*(x)$ in our method is $-\log D_w^*(x) = \log\left((p + p_\theta)/((1 - r)p + p_\theta)\right)$. In cases where imperfect demonstrations exhibit a pronounced preference for a certain state-action pair, resulting in a significantly higher probability $p$ compared to other state-action pairs, the presence of the term $(1 - r)p$ in the denominator mitigates the impact of an excessively high $p$. Furthermore, even in extreme scenarios where $p$ is much greater than $p_\theta$, the maximum reward provided by the discriminator in our method is $-log(1-r)$, rather than approaching positive infinity as in 2IWIL. As a result, in PN-GAIL, the discriminator can offer more accurate rewards, thereby facilitating the subsequent RL process to learn a better policy.

In the following theorem, we demonstrate that the estimation error of Eq. (13) is bounded, indicating that we can obtain a classifier by minimizing $\widehat{R}_{\text{BSC},\ell}$. We provide the estimation error bound with Rademacher complexity (Bartlett & Mendelson, 2002).

**Theorem 4.4.** *Denote $\mathcal{G}$ as the hypothesis class being utilized and $\mathfrak{R}_n(\mathcal{G})$ as the Rademacher complexity of the function class $\mathcal{G}$ with a sample size of $n$. Assume that the loss function $\ell$ is $\rho_\ell$-Lipschitz continuous, and there exists a constant $C_\ell > 0$ such that for any $g \in \mathcal{G}$, $\sup_{x \in \mathcal{X}, y \in \{\pm 1\}} |\ell(yg(x))| \leq C_\ell$. Define $\hat{g}$ as the minimizer of $\widehat{R}_{\text{BSC},\ell}(g)$ over $g \in \mathcal{G}$ and $g^*$ as the minimizer of $R_{\text{BSC},\ell}(g)$ over $g \in \mathcal{G}$. For $\delta \in (0, 1)$, with probability at least $1 - \delta$ when repeatedly sampling data to train $\hat{g}$, we have*

$$R_{\text{BSC},\ell}(\hat{g}) - R_{\text{BSC},\ell}(g^*) \leq 16\rho_L((3 + \alpha - \beta)\mathfrak{R}_{n_c}(\mathcal{G}) + (\alpha + \beta)\mathfrak{R}_{n_u}(\mathcal{G}))$$

$$+ 4C_L\sqrt{\frac{\log(12/\delta)}{2}}\left((3 + \alpha - \beta)n_c^{-\frac{1}{2}} + (\alpha + \beta)n_u^{-\frac{1}{2}}\right). \tag{16}$$

## 4.4 OVERALL ALGORITHM

Through the aforementioned classifier, we can obtain the confidence scores for all demonstrations in the unlabeled data $\mathcal{D}_u$. Subsequently, we treat both $\mathcal{D}_c$ and $\mathcal{D}_u$ as imperfect demonstrations and optimize the discriminator $D_w$. Finally, we utilize Trust Region Policy Optimization (TRPO) (Schulman et al., 2015) to learn a policy $\pi_\theta$ based on the rewards provided by the discriminator. The pseudocode for the overall algorithm can be found in Appendix A.

## 5 EXPERIMENTS

In this section, we validate our method by conducting experiments on six control tasks, including Pendulum-v1 and five challenging MuJoCo (Todorov et al., 2012) environments. We aim to answer three questions: (1) Is 2IWIL influenced by the preferences inherent in imperfect demonstrations, and can our method alleviate such influence? (2) Does our proposed BSC outperform the SC proposed in 2IWIL? (3) How robust is our method?

**Task setup** We conduct experiments across six environments (Pendulum-v1, Ant-v2, Walker2d-v2, Hopper-v2, Swimmer-v2, and HalfCheetah-v2). Each experiment is conducted using five different random seeds. Additionally, to better showcase the performance of imitation, we normalize the cumulative rewards of the policies, where $1.0$ represents the optimal policy and $0.0$ represents the random policy. Due to space constraints, we place the details of the experiments, the performance of the optimal and the random policies and the uncropped figures of Ant-v2 in Appendix C.1, C.4.

**Demonstrations** For the Pendulum-v1 environment, we train an optimal policy $\pi_{opt}$ and an intermediate policy $\pi_1$ using TRPO. To highlight the preferences inherent in imperfect demonstrations, we aim for a higher proportion of samples to be drawn from $\pi_1$. In that way, we ensure that the number of demonstrations generated by $\pi_1$ is four times that of $\pi_{opt}$, resulting in a final demonstrations ratio of $\pi_{opt} : \pi_1 = 1 : 4$, which are then merged together. Afterward, all demonstrations are annotated with confidence scores, utilizing normalized rewards. For the Ant-v2, Walker2d-v2, Hopper-v2, Swimmer-v2, and HalfCheetah-v2 environments, to maintain fairness, we directly utilize the demonstrations and confidence scores provided by the code of 2IWIL. During the practical experiments across all six environments, $20\%$ of the given demonstrations are randomly selected to be assigned confidence scores, which means that the label ratio is $0.2$.

**Baselines** We choose GAIL, 2IWIL, IC-GAIL, and WGAIL as our baseline methods. Among these methods, since GAIL and WGAIL do not require confidence information, we only provide them with demonstrations. Furthermore, we conduct ablation experiments, including **2IWIL**: Original 2IWIL. **PN-GAIL\BSC**: PN-GAIL with *semi-conf (SC) classification*. **PN-GAIL\PN**: 2IWIL with *balanced semi-conf (BSC) classification*. **PN-GAIL**: Our final method. All methods are trained jointly using both $\mathcal{D}_c$ and $\mathcal{D}_u$. Meanwhile, we also test the performance of CAIL, ranking-based methods (T-REX, D-REX) and f-IRL (Ni et al., 2021) by constructing trajectory rankings from confidence scores in the Pendulum-v1 and Ant-v2 environments (due to the demonstrations provided by the 2IWIL's code is not in trajectory form). The results can be seen in Appendix C.3.

### 5.1 PERFORMANCE

In our experiments, we use different numbers of $\mathcal{D}_c + \mathcal{D}_u$ for different tasks, and the specific values are shown in Appendix C.1. Fig. 2 and Fig. 3 show the normalized average returns during training. The results in Fig. 2 demonstrate that our method outperforms other baseline methods, achieving the highest returns in all six environments. Of particular note is its performance in Pendulum-v1. Here, imperfect demonstrations exhibit a preference for certain state-action pairs with lower confidence scores, adversely affecting the learning process of 2IWIL and leading to a poor policy. In contrast, our method addresses this issue by incorporating the negative risk of imperfect demonstrations. Experimental results demonstrate that our method is able to learn a near-optimal policy in the Pendulum-v1 environment while other baseline methods fail.

We observe that the performance of GAIL generally falls below that of other methods. This is because GAIL treats all demonstrations as optimal, unable to allocate distinct weights to different

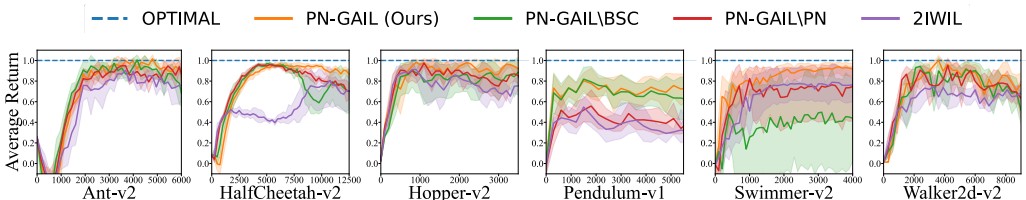

Figure 2: Normalized average returns of PN-GAIL and baseline methods during training. The x-axis is the number of training steps.

Figure 3: Normalized average returns of ablation experiments during training. The x-axis is the number of training steps.

demonstrations. However, in Walker2d-v2, neither 2IWIL nor IC-GAIL outperforms GAIL. We feel this might be due to the relatively low average confidence of demonstrations in Walker2d-v2. Meanwhile, we notice that WGAIL performs worse than GAIL in Walker2d-v2, which we attribute to its assumption of a higher proportion of optimal demonstrations within the imperfect demonstrations. Since the demonstrations provided in Walker2d-v2 do not align with this assumption, the confidence estimation of WGAIL would no longer be accurate.

Additionally, Fig. 3 shows the normalized average returns of the ablation experiments. In Fig. 3, the large difference between the performance of PN-GAIL and PN-GAIL\PN indicates that there is a preference in the imperfect demonstrations, resulting in the poor performance of the 2IWIL follow-up method. The large performance gap between the performance of PN-GAIL and PN-GAIL\BSC indicates that the prediction confidence of SC classification is not accurate enough, which affects the subsequent training. If the performance gap is not significant, it means that the above problems are not obvious or do not affect the final results. Our method outperforms other methods across all environments, thus confirming the performance enhancement brought by incorporating the negative risk of imperfect demonstrations and employing *balanced semi-conf (BSC) classification*.

## 5.2 ACCURACY OF CLASSIFIER

By comparing PN-GAIL with PN-GAIL\BSC as depicted in Fig. 3, it is clear that the performance of PN-GAIL can be improved by using the BSC classifier. This observation demonstrates the superior capability of the BSC classifier over the SC classifier in accurately predicting confidence scores. To illustrate the disparity between these two classifiers more clearly, we calculate the Mean Absolute Error (MAE) and Root Mean Square Error (RMSE) of the prediction confidence scores. Here, MAE represents the average of absolute errors, while RMSE denotes the square root of the average of squared differences between predicted and true values. As shown in Table 1, the MAE and RMSE of the BSC classifier are notably lower than those of the SC classifier, indicating that the predictions of the BSC classifier are closer to the ground truth. Consequently, BSC classifier provides more accurate confidence scores for subsequent imitation learning.

## 5.3 ROBUSTNESS OF PN-GAIL

To test the robustness of our method, We evaluate the performance of PN-GAIL at different label ratios in Ant-v2 and Hopper-v2 environments, the results are shown in Fig. 4 (a) and (b). As the label ratio decreases, PN-GAIL exhibits only a marginal decline in performance. This indicates that

Table 1: Accuracy of classifier measured by MAE and RMSE.

| Classifier | Metrics | Ant-v2 | HalfCheetah-v2 | Hopper-v2 | Pendulum-v1 | Swimmer-v2 | Walker2d-v2 |
|---|---|---|---|---|---|---|---|
| SC | MAE | $0.213 \pm 0.023$ | $0.184 \pm 0.011$ | $0.307 \pm 0.025$ | $0.126 \pm 0.014$ | $0.362 \pm 0.049$ | $0.132 \pm 0.015$ |
| | RMSE | $0.345 \pm 0.033$ | $0.272 \pm 0.009$ | $0.519 \pm 0.022$ | $0.164 \pm 0.013$ | $0.595 \pm 0.040$ | $0.246 \pm 0.032$ |
| BSC | MAE | $\mathbf{0.056 \pm 0.011}$ | $\mathbf{0.057 \pm 0.012}$ | $\mathbf{0.169 \pm 0.126}$ | $\mathbf{0.097 \pm 0.006}$ | $\mathbf{0.286 \pm 0.179}$ | $\mathbf{0.014 \pm 0.002}$ |
| | RMSE | $\mathbf{0.212 \pm 0.026}$ | $\mathbf{0.175 \pm 0.013}$ | $\mathbf{0.371 \pm 0.138}$ | $\mathbf{0.138 \pm 0.005}$ | $\mathbf{0.472 \pm 0.188}$ | $\mathbf{0.101 \pm 0.010}$ |

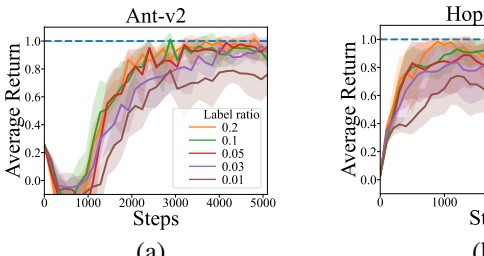
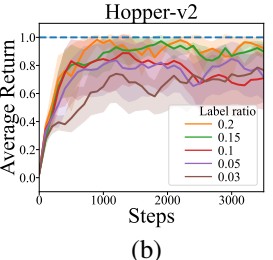
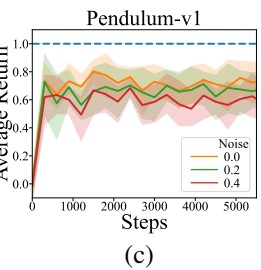

(a)   (b)   (c)

Figure 4: (a) Ant-v2 experiments with different label ratios. (b) Hopper-v2 experiments with different label ratios. (c) Pendulum-v1 experiments with different standard deviations of Gaussian noise.

PN-GAIL is not highly dependent on the label ratio, maintaining excellent performance even as the label ratio decreases.

In practice, considering that confidence scores are typically provided by human annotators, variations in their standards for labeling confidence may arise due to individual differences and factors such as fatigue. To assess the robustness of our method against noise in confidence scores, we conduct additional experiments. In Pendulum-v1, we introduce Gaussian noise to the confidence scores: $\hat{r}(x) = \text{clip}_{[0,1]}(r(x) + \epsilon)$, where $\epsilon \sim \mathcal{N}(0, \sigma^2)$, $\text{clip}_{[l,u]}(v) = \min\{\max\{v, l\}, u\}$. As shown in Fig. 4 (c), the numbers indicate the standard deviations of Gaussian noise. Even when confidence scores are subject to noise, our method still demonstrates satisfactory performance, indicating its robustness to noisy confidence scores.

We also test the performance of PN-GAIL in two scenarios: first, by reducing the number of unlabeled demonstrations; and second, by observing how PN-GAIL performs when the average optimality of imperfect demonstrations changes. Due to space constraints, we present the details of these experiments and the corresponding figures in Appendix C.3.

## 6   CONCLUSION

In this work, we proposed a novel algorithm termed PN-GAIL for imitation learning from imperfect demonstrations. PN-GAIL leverages non-optimal information embedded in these demonstrations, enabling the discriminator to weigh both positive and negative risks in a holistic manner. This approach facilitates the assignment of more refined reward signals. To enhance the precision of confidence estimation, we have integrated an advanced semi-supervised confidence classifier into our framework. Our theoretical investigations demonstrate that PN-GAIL is not merely capable of mimicking imperfect demonstrations but also adept at circumventing the imitation of suboptimal behaviors, thereby ensuring the acquisition of an optimal policy. Comprehensive experimental results indicate that our approach surpasses existing baselines in performance and exhibits remarkable robustness, thereby establishing a robust foundation for the practical deployment of imitation learning in real-world scenarios.

ACKNOWLEDGEMENT

This work was supported in part by the National Natural Science Foundation of China (Nos. 72394363 & 62073160), the Nanjing University Integrated Research Platform of the Ministry of Education-Top Talents Program and the Australian Research Council (FT220100656).

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

# Appendices

## A  ALGORITHM

---
**Algorithm 1** PN-GAIL
---
1: **Input:** Imperfect demonstrations and confidence $\mathcal{D}_{\mathrm{c}} = \{(x_{\mathrm{c},i}, r_i)\}_{i=1}^{n_{\mathrm{c}}}, \mathcal{D}_{\mathrm{u}} = \{x_{\mathrm{u},i}\}_{i=1}^{n_{\mathrm{u}}}$
2: Train a probabilistic classifier by minimizing Eq. (13) with $\alpha = \frac{n_{\mathrm{u}}}{n_{\mathrm{c}}+n_{\mathrm{u}}}, \beta = \frac{n_{\mathrm{u}}}{n_{\mathrm{c}}+n_{\mathrm{u}}}$
3: Predict confidence scores $\{\hat{r}_{\mathrm{u},i}\}_{i=1}^{n_{\mathrm{u}}}$ for $\{x_{\mathrm{u},i}\}_{i=1}^{n_{\mathrm{u}}}$
4: **for** $i = 0, 1, 2, ...$ **do**
5:    Sample trajectories $\tau_\theta \sim \pi_\theta, \tau_e \sim \{\mathcal{D}_{\mathrm{c}}, \mathcal{D}_{\mathrm{u}}\}$
6:    Update $D_w$ by maximizing Eq. (11)
7:    Update $\pi_\theta$ by TRPO with reward $-\log D_w(x)$
8: **end for**

---

## B  DERIVATIONS AND PROOFS

### B.1  PROOF OF THEOREM 4.1

**Theorem.** *Based on Eq. (9), the objective of the discriminator can be rewritten as*

$$\max_w \mathbb{E}_{x \sim p_\theta}\left[\log D_w(x)\right] + \mathbb{E}_{x \sim p}\left[r(x)\log(1 - D_w(x))\right] + \mathbb{E}_{x \sim p}\left[(1 - r(x))\log D_w(x)\right].$$

*Proof.* Since $p_{\mathrm{opt}}(x) = p(x|y=0) = \frac{r(x)p(x)}{\eta}, p_{\mathrm{non}}(x) = p(x|y=1) = \frac{(1-r(x))p(x)}{1-\eta}$, we have

$$
\begin{aligned}
\eta \mathbb{E}_{x \sim p_{\mathrm{opt}}}[\log(1 - D_w(x))] &= \eta \int p_{\mathrm{opt}}(x)\log(1 - D_w(x))dx \\
&= \eta \int \frac{r(x)}{\eta}p(x)\log(1 - D_w(x))dx \\
&= \int p(x)r(x)\log(1 - D_w(x))dx \\
&= \mathbb{E}_{x \sim p}\left[r(x)\log(1 - D_w(x))\right],
\end{aligned}
$$

$$
\begin{aligned}
(1-\eta)\mathbb{E}_{x \sim p_{\mathrm{non}}}[\log D_w(x)] &= (1-\eta) \int p_{\mathrm{non}}(x)\log D_w(x)dx \\
&= (1-\eta) \int \frac{1 - r(x)}{1 - \eta}p(x)\log D_w(x)dx \\
&= \int p(x)(1 - r(x))\log D_w(x)dx \\
&= \mathbb{E}_{x \sim p}\left[(1 - r(x))\log D_w(x)\right],
\end{aligned}
$$

According to Eq. (8), the objective of the discriminator can be rewritten as

$$\max_w \mathbb{E}_{x \sim p_\theta}\left[\log D_w(x)\right] + \mathbb{E}_{x \sim p}\left[r(x)\log(1 - D_w(x))\right] + \mathbb{E}_{x \sim p}\left[(1 - r(x))\log D_w(x)\right].$$

$\square$

### B.2  DERIVATION OF BALANCED SEMI-CONF (BSC) CLASSIFICATION

Recall that in 2IWIL:

$$R_{\mathrm{SC},\ell}(g) = \mathbb{E}_{x,r \sim q}[r\ell(g(x)) + (1 - r)\ell(-g(x))]$$

$$= \mathbb{E}_{x,r \sim q}\left[r\ell(g(x)) + (1 - r)\ell(-g(x)) + \underbrace{\beta\ell(-g(x)) - \beta\ell(-g(x))}_{=0}\right]$$

By introducing $\mathbb{E}_{x\sim p}[\alpha\ell(g(x))] - \mathbb{E}_{x\sim q}[\alpha\ell(g(x))]$ with theoretical values of 0, we have

$$
R_{\text{BSC},\ell}(g) = \mathbb{E}_{x,r\sim q}\left[r\ell(g(x)) + (1-r)\ell(-g(x)) + \underbrace{\alpha\ell(g(x)) - \alpha\ell(g(x))}_{=0} + \underbrace{\beta\ell(-g(x)) - \beta\ell(-g(x))}_{=0}\right]
$$
$$
= \mathbb{E}_{x,r\sim q}\left[r\ell(g(x)) + (1-r)\ell(-g(x)) - \alpha\ell(g(x)) - \beta\ell(-g(x))\right] + \mathbb{E}_{x\sim p}[\alpha\ell(g(x)) + \beta\ell(-g(x))].
$$

### B.3 PROOF OF THEOREM 4.2

**Theorem.** *Let $d_1$ denote* $\text{Var}(\ell(-g(x)))$, $d_2$ *denote* $\text{Var}(\ell(g(x)))$, $\sigma_{\text{cov1}}$ *denote the covariance between* $\frac{1}{n_c}\sum_{i=1}^{n_c} r_i(\ell(g(x_{c,i})) - \ell(-g(x_{c,i})))$ *and* $\frac{1}{n_c}\sum_{i=1}^{n_c}\ell(-g(x_{c,i}))$, $\sigma_{\text{cov2}}$ *denote the covariance between* $\frac{1}{n_c}\sum_{i=1}^{n_c}(1-r_i)(\ell(-g(x_{c,i})) - \ell(g(x_{c,i})))$ *and* $\frac{1}{n_c}\sum_{i=1}^{n_c}\ell(g(x_{c,i}))$, *cov denote* $\text{Cov}(\ell(-g(x)), \ell(g(x)))$. *The estimator* $\widehat{R}_{\text{BSC},\ell}(g)$ *has the minimum variance when*

$$
\alpha = \frac{n_u}{n_c + n_u} - \frac{d_1 cov - cov^2}{d_1 d_2 - cov^2}\frac{n_u}{n_c + n_u} + \frac{d_1\sigma_{\text{cov2}} - cov\sigma_{\text{cov1}}}{d_1 d_2 - cov^2}\frac{n_c n_u}{n_c + n_u},
$$
$$
\beta = \frac{n_u}{n_c + n_u} - \frac{d_2 cov - cov^2}{d_1 d_2 - cov^2}\frac{n_u}{n_c + n_u} + \frac{d_2\sigma_{\text{cov1}} - cov\sigma_{\text{cov2}}}{d_1 d_2 - cov^2}\frac{n_c n_u}{n_c + n_u}.
$$

*Proof.* Denote

$$
\mu \triangleq \mathbb{E}_{\mathcal{D}_c,\mathcal{D}_u}\left[\widehat{R}_{\text{BSC},\ell}(g)\right],
$$
$$
\mu_1 \triangleq \mathbb{E}_{\mathcal{D}_c}\left[\frac{1}{n_c}\sum_{i=1}^{n_c}\ell(-g(x_{c,i}))\right] = \mathbb{E}_{\mathcal{D}_u}\left[\frac{1}{n_u}\sum_{i=1}^{n_u}\ell(-g(x_{u,i}))\right] = \mathbb{E}_{x\sim p}[\ell(-g(x))],
$$
$$
\mu_2 \triangleq \mathbb{E}_{\mathcal{D}_c}\left[\frac{1}{n_c}\sum_{i=1}^{n_c}\ell(g(x_{c,i}))\right] = \mathbb{E}_{\mathcal{D}_u}\left[\frac{1}{n_u}\sum_{i=1}^{n_u}\ell(g(x_{u,i}))\right] = \mathbb{E}_{x\sim p}[\ell(g(x))],
$$
$$
d_1 \triangleq \text{Var}_{\mathcal{D}_c}[\ell(-g(x_c))] = \text{Var}_{\mathcal{D}_u}[\ell(-g(x_u))] = \text{Var}[\ell(-g(x))],
$$
$$
d_2 \triangleq \text{Var}_{\mathcal{D}_c}[\ell(g(x_c))] = \text{Var}_{\mathcal{D}_u}[\ell(g(x_u))] = \text{Var}[\ell(g(x))],
$$
$$
\omega \triangleq \mathbb{E}_{\mathcal{D}_c}\left[\frac{1}{n_c}\sum_{i=1}^{n_c}\ell(-g(x_{c,i}))\ell(g(x_{c,i}))\right] = \mathbb{E}_{\mathcal{D}_u}\left[\frac{1}{n_u}\sum_{i=1}^{n_u}\ell(-g(x_{u,i}))\ell(g(x_{u,i}))\right] = \mathbb{E}_{x\sim p}[\ell(-g(x))\ell(g(x))],
$$
$$
cov \triangleq \text{Cov}_{\mathcal{D}_c}(\ell(-g(x_c)), \ell(g(x_c))) = \text{Cov}_{\mathcal{D}_u}(\ell(-g(x_u)), \ell(g(x_u))) = \text{Cov}(\ell(-g(x)), \ell(g(x))) = \omega - \mu_1\mu_2,
$$
$$
\sigma_{\text{cov1}} \triangleq \text{Cov}\left(\frac{1}{n_c}\sum_{i=1}^{n_c} r_i(\ell(g(x_{c,i})) - \ell(-g(x_{c,i}))), \frac{1}{n_c}\sum_{i=1}^{n_c}\ell(-g(x_{c,i}))\right),
$$
$$
\sigma_{\text{cov2}} \triangleq \text{Cov}\left(\frac{1}{n_c}\sum_{i=1}^{n_c}(1-r_i)(\ell(-g(x_{c,i})) - \ell(g(x_{c,i}))), \frac{1}{n_c}\sum_{i=1}^{n_c}\ell(g(x_{c,i}))\right).
$$

Next, we adopt the symbols defined above to express several formulas that will be used:

$$
\mathbb{E}_{\mathcal{D}_c}\left[\left(\frac{1}{n_c}\sum_{i=1}^{n_c}\ell(-g(x_{c,i}))\right)^2\right] = \frac{1}{n_c^2}\mathbb{E}_{\mathcal{D}_c}\left[\sum_{i=1}^{n_c}\ell(-g(x_{c,i}))^2 + 2\sum_{i=1}^{n_c}\sum_{j=1}^{i-1}\ell(-g(x_{c,i}))\ell(-g(x_{c,j}))\right]
$$
$$
= \frac{1}{n_c^2}\left(n_c\mathbb{E}_{x\sim p}\left[\ell(-g(x))^2\right] + n_c(n_c-1)\mathbb{E}_{x\sim p}\left[\ell(-g(x))\right]^2\right)
$$
$$
= \frac{1}{n_c}\text{Var}(\ell(-g(x))) + \mu_1^2
$$
$$
= \frac{1}{n_c}d_1 + \mu_1^2. \tag{17}
$$

Similarly, we obtain

$$\mathbb{E}_{\mathcal{D}_{\mathrm{u}}}\left[\left(\frac{1}{n_{\mathrm{u}}}\sum_{i=1}^{n_{\mathrm{u}}}\ell(-g(x_{\mathrm{u},i}))\right)^2\right] = \frac{1}{n_{\mathrm{u}}}d_1 + \mu_1^2, \tag{18}$$

$$\mathbb{E}_{\mathcal{D}_{\mathrm{c}}}\left[\left(\frac{1}{n_{\mathrm{c}}}\sum_{i=1}^{n_{\mathrm{c}}}\ell(g(x_{\mathrm{c},i}))\right)^2\right] = \frac{1}{n_{\mathrm{c}}}d_2 + \mu_2^2, \tag{19}$$

$$\mathbb{E}_{\mathcal{D}_{\mathrm{u}}}\left[\left(\frac{1}{n_{\mathrm{u}}}\sum_{i=1}^{n_{\mathrm{u}}}\ell(g(x_{\mathrm{u},i}))\right)^2\right] = \frac{1}{n_{\mathrm{u}}}d_2 + \mu_2^2, \tag{20}$$

$$\mathbb{E}_{\mathcal{D}_{\mathrm{u}}}\left[\left(\frac{1}{n_{\mathrm{u}}}\sum_{i=1}^{n_{\mathrm{u}}}\ell(-g(x_{\mathrm{u},i}))\right)\left(\frac{1}{n_{\mathrm{u}}}\sum_{i=1}^{n_{\mathrm{u}}}\ell(g(x_{\mathrm{u},i}))\right)\right] = \frac{1}{n_{\mathrm{u}}}cov + \mu_1\mu_2, \tag{21}$$

$$\mathbb{E}_{\mathcal{D}_{\mathrm{c}}}\left[\left(\frac{1}{n_{\mathrm{c}}}\sum_{i=1}^{n_{\mathrm{c}}}\ell(-g(x_{\mathrm{c},i}))\right)\left(\frac{1}{n_{\mathrm{c}}}\sum_{i=1}^{n_{\mathrm{c}}}\ell(g(x_{\mathrm{c},i}))\right)\right] = \frac{1}{n_{\mathrm{c}}}cov + \mu_1\mu_2. \tag{22}$$

We also have

$$\mathbb{E}_{\mathcal{D}_{\mathrm{c}},\mathcal{D}_{\mathrm{u}}}\left[\left(\frac{1}{n_{\mathrm{c}}}\sum_{i=1}^{n_{\mathrm{c}}}r(x_i)(\ell(g(x_{\mathrm{c},i})) - \ell(-g(x_{\mathrm{c},i})))\right)\left(\frac{1}{n_{\mathrm{u}}}\sum_{i=1}^{n_{\mathrm{u}}}\ell(-g(x_{\mathrm{u},i})) - \frac{1}{n_{\mathrm{c}}}\sum_{i=1}^{n_{\mathrm{c}}}\ell(-g(x_{\mathrm{c},i}))\right)\right]$$

$$= \mathbb{E}_{\mathcal{D}_{\mathrm{c}}}\left[\left(\frac{1}{n_{\mathrm{c}}}\sum_{i=1}^{n_{\mathrm{c}}}r(x_i)(\ell(g(x_{\mathrm{c},i})) - \ell(-g(x_{\mathrm{c},i})))\right)\right]\mathbb{E}_{\mathcal{D}_{\mathrm{u}}}\left[\left(\frac{1}{n_{\mathrm{u}}}\sum_{i=1}^{n_{\mathrm{u}}}\ell(-g(x_{\mathrm{u},i}))\right)\right]$$

$$\quad - \mathbb{E}_{\mathcal{D}_{\mathrm{c}}}\left[\left(\frac{1}{n_{\mathrm{c}}}\sum_{i=1}^{n_{\mathrm{c}}}r(x_i)(\ell(g(x_{\mathrm{c},i})) - \ell(-g(x_{\mathrm{c},i})))\right)\left(\frac{1}{n_{\mathrm{c}}}\sum_{i=1}^{n_{\mathrm{c}}}\ell(-g(x_{\mathrm{c},i}))\right)\right]$$

$$= \mathbb{E}_{\mathcal{D}_{\mathrm{c}}}\left[\left(\frac{1}{n_{\mathrm{c}}}\sum_{i=1}^{n_{\mathrm{c}}}r(x_i)(\ell(g(x_{\mathrm{c},i})) - \ell(-g(x_{\mathrm{c},i})))\right)\right]\mathbb{E}_{\mathcal{D}_{\mathrm{c}}}\left[\left(\frac{1}{n_{\mathrm{c}}}\sum_{i=1}^{n_{\mathrm{c}}}\ell(-g(x_{\mathrm{c},i}))\right)\right]$$

$$\quad - \mathbb{E}_{\mathcal{D}_{\mathrm{c}}}\left[\left(\frac{1}{n_{\mathrm{c}}}\sum_{i=1}^{n_{\mathrm{c}}}r(x_i)(\ell(g(x_{\mathrm{c},i})) - \ell(-g(x_{\mathrm{c},i})))\right)\left(\frac{1}{n_{\mathrm{c}}}\sum_{i=1}^{n_{\mathrm{c}}}\ell(-g(x_{\mathrm{c},i}))\right)\right]$$

$$= \mathrm{Cov}\left(\frac{1}{n_{\mathrm{c}}}\sum_{i=1}^{n_{\mathrm{c}}}r_i\left(\ell\left(g\left(x_{\mathrm{c},i}\right)\right) - \ell\left(-g\left(x_{\mathrm{c},i}\right)\right)\right), \frac{1}{n_{\mathrm{c}}}\sum_{i=1}^{n_{\mathrm{c}}}\ell\left(-g\left(x_{\mathrm{c},i}\right)\right)\right)$$

$$= -\sigma_{\mathrm{cov1}}. \tag{23}$$

Similarly, we obtain

$$\mathbb{E}_{\mathcal{D}_{\mathrm{c}},\mathcal{D}_{\mathrm{u}}}\left[\left(\frac{1}{n_{\mathrm{c}}}\sum_{i=1}^{n_{\mathrm{c}}}(1 - r(x_i))(\ell(-g(x_{\mathrm{c},i})) - \ell(g(x_{\mathrm{c},i})))\right)\left(\frac{1}{n_{\mathrm{u}}}\sum_{i=1}^{n_{\mathrm{u}}}\ell(g(x_{\mathrm{u},i})) - \frac{1}{n_{\mathrm{c}}}\sum_{i=1}^{n_{\mathrm{c}}}\ell(g(x_{\mathrm{c},i}))\right)\right]$$

$$= -\sigma_{\mathrm{cov2}}. \tag{24}$$

Hence, we can express the estimator variance $\mathrm{Var}(\widehat{R}_{\mathrm{BSC},\ell}(g))$ as follows:

$\mathrm{Var}(\widehat{R}_{\mathrm{BSC},\ell}(g))$

$= \mathbb{E}_{\mathcal{D}_{\mathrm{c}},\mathcal{D}_{\mathrm{u}}}\left[\left(\widehat{R}_{\mathrm{BSC},\ell}(g)\right)^2\right] - \mu^2$

$= \mathbb{E}_{\mathcal{D}_{\mathrm{c}},\mathcal{D}_{\mathrm{u}}}\left[\left(\underbrace{\frac{1}{n_{\mathrm{c}}}\sum_{i=1}^{n_{\mathrm{c}}}r_i\ell(g(x_{\mathrm{c},i})) + (1-r_i)\ell(-g(x_{\mathrm{c},i}))}_{A} + \underbrace{\beta(\frac{1}{n_{\mathrm{u}}}\sum_{i=1}^{n_{\mathrm{u}}}\ell(-g(x_{\mathrm{u},i})) - \frac{1}{n_{\mathrm{c}}}\sum_{i=1}^{n_{\mathrm{c}}}\ell(-g(x_{\mathrm{c},i})))}_{B}\right.\right.$

$\left.\left. + \underbrace{\alpha(\frac{1}{n_{\mathrm{u}}}\sum_{i=1}^{n_{\mathrm{u}}}\ell(g(x_{\mathrm{u},i})) - \frac{1}{n_{\mathrm{c}}}\sum_{i=1}^{n_{\mathrm{c}}}\ell(g(x_{\mathrm{c},i})))}_{C}\right)^2\right] - \mu^2$

$= \mathbb{E}_{\mathcal{D}_{\mathrm{c}}}[A^2] + \beta^2\mathbb{E}_{\mathcal{D}_{\mathrm{c}},\mathcal{D}_{\mathrm{u}}}[B^2] + \alpha^2\mathbb{E}_{\mathcal{D}_{\mathrm{c}},\mathcal{D}_{\mathrm{u}}}[C^2] + 2\beta\mathbb{E}_{\mathcal{D}_{\mathrm{c}},\mathcal{D}_{\mathrm{u}}}[AB] + 2\alpha\mathbb{E}_{\mathcal{D}_{\mathrm{c}},\mathcal{D}_{\mathrm{u}}}[AC] + 2\beta\alpha\mathbb{E}_{\mathcal{D}_{\mathrm{c}},\mathcal{D}_{\mathrm{u}}}[BC] - \mu^2$

$= \mathbb{E}_{\mathcal{D}_{\mathrm{c}},\mathcal{D}_{\mathrm{u}}}[B^2]\left(\beta + \frac{\mathbb{E}_{\mathcal{D}_{\mathrm{c}},\mathcal{D}_{\mathrm{u}}}[AB]}{\mathbb{E}_{\mathcal{D}_{\mathrm{c}},\mathcal{D}_{\mathrm{u}}}[B^2]} + \frac{\mathbb{E}_{\mathcal{D}_{\mathrm{c}},\mathcal{D}_{\mathrm{u}}}[BC]}{\mathbb{E}_{\mathcal{D}_{\mathrm{c}},\mathcal{D}_{\mathrm{u}}}[B^2]}\alpha\right)^2 + \left(\mathbb{E}_{\mathcal{D}_{\mathrm{c}},\mathcal{D}_{\mathrm{u}}}[C^2] - \frac{\mathbb{E}^2_{\mathcal{D}_{\mathrm{c}},\mathcal{D}_{\mathrm{u}}}[BC]}{\mathbb{E}_{\mathcal{D}_{\mathrm{c}},\mathcal{D}_{\mathrm{u}}}[B^2]}\right)\alpha^2$

$\quad + 2\left(\mathbb{E}_{\mathcal{D}_{\mathrm{c}},\mathcal{D}_{\mathrm{u}}}[AC] - \frac{\mathbb{E}_{\mathcal{D}_{\mathrm{c}},\mathcal{D}_{\mathrm{u}}}[AB]\mathbb{E}_{\mathcal{D}_{\mathrm{c}},\mathcal{D}_{\mathrm{u}}}[BC]}{\mathbb{E}_{\mathcal{D}_{\mathrm{c}},\mathcal{D}_{\mathrm{u}}}[B^2]}\right)\alpha + \underbrace{\mathbb{E}_{\mathcal{D}_{\mathrm{c}},\mathcal{D}_{\mathrm{u}}}[A^2] - \frac{\mathbb{E}^2_{\mathcal{D}_{\mathrm{c}},\mathcal{D}_{\mathrm{u}}}[AB]}{\mathbb{E}_{\mathcal{D}_{\mathrm{c}},\mathcal{D}_{\mathrm{u}}}[B^2]} - \mu^2}_{\mathrm{const.w.r.t.}\alpha,\beta}$

$= \mathbb{E}_{\mathcal{D}_{\mathrm{c}},\mathcal{D}_{\mathrm{u}}}[B^2]\left(\beta + \frac{\mathbb{E}_{\mathcal{D}_{\mathrm{c}},\mathcal{D}_{\mathrm{u}}}[AB]}{\mathbb{E}_{\mathcal{D}_{\mathrm{c}},\mathcal{D}_{\mathrm{u}}}[B^2]} + \frac{\mathbb{E}_{\mathcal{D}_{\mathrm{c}},\mathcal{D}_{\mathrm{u}}}[BC]}{\mathbb{E}_{\mathcal{D}_{\mathrm{c}},\mathcal{D}_{\mathrm{u}}}[B^2]}\alpha\right)^2$

$\quad + \left(\mathbb{E}_{\mathcal{D}_{\mathrm{c}},\mathcal{D}_{\mathrm{u}}}[C^2] - \frac{\mathbb{E}^2_{\mathcal{D}_{\mathrm{c}},\mathcal{D}_{\mathrm{u}}}[BC]}{\mathbb{E}_{\mathcal{D}_{\mathrm{c}},\mathcal{D}_{\mathrm{u}}}[B^2]}\right)\left(\alpha + \frac{\mathbb{E}_{\mathcal{D}_{\mathrm{c}},\mathcal{D}_{\mathrm{u}}}[AC]\mathbb{E}_{\mathcal{D}_{\mathrm{c}},\mathcal{D}_{\mathrm{u}}}[B^2] - \mathbb{E}_{\mathcal{D}_{\mathrm{c}},\mathcal{D}_{\mathrm{u}}}[AB]\mathbb{E}_{\mathcal{D}_{\mathrm{c}},\mathcal{D}_{\mathrm{u}}}[BC]}{\mathbb{E}_{\mathcal{D}_{\mathrm{c}},\mathcal{D}_{\mathrm{u}}}[B^2]\mathbb{E}_{\mathcal{D}_{\mathrm{c}},\mathcal{D}_{\mathrm{u}}}[C^2] - \mathbb{E}^2_{\mathcal{D}_{\mathrm{c}},\mathcal{D}_{\mathrm{u}}}[BC]}\right)^2$

$\quad \underbrace{- \left(\mathbb{E}_{\mathcal{D}_{\mathrm{c}},\mathcal{D}_{\mathrm{u}}}[C^2] - \frac{\mathbb{E}^2_{\mathcal{D}_{\mathrm{c}},\mathcal{D}_{\mathrm{u}}}[BC]}{\mathbb{E}_{\mathcal{D}_{\mathrm{c}},\mathcal{D}_{\mathrm{u}}}[B^2]}\right)\left(\frac{\mathbb{E}_{\mathcal{D}_{\mathrm{c}},\mathcal{D}_{\mathrm{u}}}[AC]\mathbb{E}_{\mathcal{D}_{\mathrm{c}},\mathcal{D}_{\mathrm{u}}}[B^2] - \mathbb{E}_{\mathcal{D}_{\mathrm{c}},\mathcal{D}_{\mathrm{u}}}[AB]\mathbb{E}_{\mathcal{D}_{\mathrm{c}},\mathcal{D}_{\mathrm{u}}}[BC]}{\mathbb{E}_{\mathcal{D}_{\mathrm{c}},\mathcal{D}_{\mathrm{u}}}[B^2]\mathbb{E}_{\mathcal{D}_{\mathrm{c}},\mathcal{D}_{\mathrm{u}}}[C^2] - \mathbb{E}^2_{\mathcal{D}_{\mathrm{c}},\mathcal{D}_{\mathrm{u}}}[BC]}\right)^2}_{\mathrm{const.w.r.t.}\alpha,\beta} + \mathrm{const}$

$= \mathbb{E}_{\mathcal{D}_{\mathrm{c}},\mathcal{D}_{\mathrm{u}}}[B^2]\left(\beta + \frac{\mathbb{E}_{\mathcal{D}_{\mathrm{c}},\mathcal{D}_{\mathrm{u}}}[AB]}{\mathbb{E}_{\mathcal{D}_{\mathrm{c}},\mathcal{D}_{\mathrm{u}}}[B^2]} + \frac{\mathbb{E}_{\mathcal{D}_{\mathrm{c}},\mathcal{D}_{\mathrm{u}}}[BC]}{\mathbb{E}_{\mathcal{D}_{\mathrm{c}},\mathcal{D}_{\mathrm{u}}}[B^2]}\alpha\right)^2$

$\quad + \left(\mathbb{E}_{\mathcal{D}_{\mathrm{c}},\mathcal{D}_{\mathrm{u}}}[C^2] - \frac{\mathbb{E}^2_{\mathcal{D}_{\mathrm{c}},\mathcal{D}_{\mathrm{u}}}[BC]}{\mathbb{E}_{\mathcal{D}_{\mathrm{c}},\mathcal{D}_{\mathrm{u}}}[B^2]}\right)\left(\alpha - \frac{\mathbb{E}_{\mathcal{D}_{\mathrm{c}},\mathcal{D}_{\mathrm{u}}}[AB]\mathbb{E}_{\mathcal{D}_{\mathrm{c}},\mathcal{D}_{\mathrm{u}}}[BC] - \mathbb{E}_{\mathcal{D}_{\mathrm{c}},\mathcal{D}_{\mathrm{u}}}[AC]\mathbb{E}_{\mathcal{D}_{\mathrm{c}},\mathcal{D}_{\mathrm{u}}}[B^2]}{\mathbb{E}_{\mathcal{D}_{\mathrm{c}},\mathcal{D}_{\mathrm{u}}}[B^2]\mathbb{E}_{\mathcal{D}_{\mathrm{c}},\mathcal{D}_{\mathrm{u}}}[C^2] - \mathbb{E}^2_{\mathcal{D}_{\mathrm{c}},\mathcal{D}_{\mathrm{u}}}[BC]}\right)^2 + \mathrm{const.}$

$$(25)$$

By Eq. (21) and Eq. (22), we can obtain

$\mathbb{E}_{\mathcal{D}_{\mathrm{c}},\mathcal{D}_{\mathrm{u}}}[BC] = \mathbb{E}_{\mathcal{D}_{\mathrm{c}},\mathcal{D}_{\mathrm{u}}}\left[\left(\frac{1}{n_{\mathrm{u}}}\sum_{i=1}^{n_{\mathrm{u}}}\ell(-g(x_{\mathrm{u},i})) - \frac{1}{n_{\mathrm{c}}}\sum_{i=1}^{n_{\mathrm{c}}}\ell(-g(x_{\mathrm{c},i}))\right)\left(\frac{1}{n_{\mathrm{u}}}\sum_{i=1}^{n_{\mathrm{u}}}\ell(g(x_{\mathrm{u},i})) - \frac{1}{n_{\mathrm{c}}}\sum_{i=1}^{n_{\mathrm{c}}}\ell(g(x_{\mathrm{c},i}))\right)\right]$

$\qquad = \frac{1}{n_{\mathrm{u}}}cov + \mu_1\mu_2 - \mu_1\mu_2 - \mu_1\mu_2 + \frac{1}{n_{\mathrm{c}}}cov + \mu_1\mu_2$

$\qquad = \left(\frac{1}{n_{\mathrm{u}}} + \frac{1}{n_{\mathrm{c}}}\right)cov.$

By Eq. (17) and Eq. (18), we can obtain

$$
\begin{aligned}
\mathbb{E}_{\mathcal{D}_{\mathrm{c}},\mathcal{D}_{\mathrm{u}}}[B^2] &= \mathbb{E}_{\mathcal{D}_{\mathrm{c}},\mathcal{D}_{\mathrm{u}}}\left[\left(\frac{1}{n_{\mathrm{u}}}\sum_{i=1}^{n_{\mathrm{u}}}\ell(-g(x_{\mathrm{u},i})) - \frac{1}{n_{\mathrm{c}}}\sum_{i=1}^{n_{\mathrm{c}}}\ell(-g(x_{\mathrm{c},i}))\right)^2\right] \\
&= \mathbb{E}_{\mathcal{D}_{\mathrm{u}}}\left[\left(\frac{1}{n_{\mathrm{u}}}\sum_{i=1}^{n_{\mathrm{u}}}\ell(-g(x_{\mathrm{u},i}))\right)^2\right] + \mathbb{E}_{\mathcal{D}_{\mathrm{c}}}\left[\left(\frac{1}{n_{\mathrm{c}}}\sum_{i=1}^{n_{\mathrm{c}}}\ell(-g(x_{\mathrm{c},i}))\right)^2\right] \\
&\quad - 2\mathbb{E}_{\mathcal{D}_{\mathrm{u}}}\left[\frac{1}{n_{\mathrm{u}}}\sum_{i=1}^{n_{\mathrm{u}}}\ell(-g(x_{\mathrm{u},i}))\right]\mathbb{E}_{\mathcal{D}_{\mathrm{c}}}\left[\frac{1}{n_{\mathrm{c}}}\sum_{i=1}^{n_{\mathrm{c}}}\ell(-g(x_{\mathrm{c},i}))\right] \\
&= \frac{1}{n_{\mathrm{u}}}d_1 + \mu_1^2 + \frac{1}{n_{\mathrm{c}}}d_1 + \mu_1^2 - 2\mu^2 \\
&= \left(\frac{1}{n_{\mathrm{u}}} + \frac{1}{n_{\mathrm{c}}}\right)d_1.
\end{aligned}
$$

Similarly, we can obtain $\mathbb{E}_{\mathcal{D}_{\mathrm{c}},\mathcal{D}_{\mathrm{u}}}[C^2] = \left(\frac{1}{n_{\mathrm{u}}} + \frac{1}{n_{\mathrm{c}}}\right)d_2$. Hence, we have $\left(\mathbb{E}_{\mathcal{D}_{\mathrm{c}},\mathcal{D}_{\mathrm{u}}}[C^2] - \frac{\mathbb{E}^2_{\mathcal{D}_{\mathrm{c}},\mathcal{D}_{\mathrm{u}}}[BC]}{\mathbb{E}_{\mathcal{D}_{\mathrm{c}},\mathcal{D}_{\mathrm{u}}}[B^2]}\right) \geq 0$ since $d_1 d_2 \geq cov^2$. By Eq. (17) and Eq. (23), we can obtain:

$$
\begin{aligned}
\mathbb{E}_{\mathcal{D}_{\mathrm{c}},\mathcal{D}_{\mathrm{u}}}[AB] &= \mathbb{E}_{\mathcal{D}_{\mathrm{c}},\mathcal{D}_{\mathrm{u}}}\left[\left(\frac{1}{n_{\mathrm{c}}}\sum_{i=1}^{n_{\mathrm{c}}}r(x_i)(\ell(g(x_{\mathrm{c},i})) - \ell(-g(x_{\mathrm{c},i}))) + \frac{1}{n_{\mathrm{c}}}\sum_{i=1}^{n_{\mathrm{c}}}\ell(-g(x_{\mathrm{c},i}))\right)\right. \\
&\quad \left.\left(\frac{1}{n_{\mathrm{u}}}\sum_{i=1}^{n_{\mathrm{u}}}\ell(-g(x_{\mathrm{u},i})) - \frac{1}{n_{\mathrm{c}}}\sum_{i=1}^{n_{\mathrm{c}}}\ell(-g(x_{\mathrm{c},i}))\right)\right] \\
&= -\sigma_{\mathrm{cov1}} + \mu_1^2 - (\frac{1}{n_{\mathrm{c}}}d_1 + \mu_1^2) \\
&= -\frac{1}{n_{\mathrm{c}}}d_1 - \sigma_{\mathrm{cov1}}.
\end{aligned}
$$

Similarly, $\mathbb{E}_{\mathcal{D}_{\mathrm{c}},\mathcal{D}_{\mathrm{u}}}[AC] = -\frac{1}{n_{\mathrm{c}}}d_2 - \sigma_{\mathrm{cov2}}$. Since $\mathbb{E}_{\mathcal{D}_{\mathrm{c}},\mathcal{D}_{\mathrm{u}}}[B^2] \geq 0$, $\left(\mathbb{E}_{\mathcal{D}_{\mathrm{c}},\mathcal{D}_{\mathrm{u}}}[C^2] - \frac{\mathbb{E}^2_{\mathcal{D}_{\mathrm{c}},\mathcal{D}_{\mathrm{u}}}[BC]}{\mathbb{E}_{\mathcal{D}_{\mathrm{c}},\mathcal{D}_{\mathrm{u}}}[B^2]}\right) \geq 0$, and $\alpha, \beta \in \mathcal{R}$, according to Eq. (25), $\mathrm{Var}(\widehat{R}_{\mathrm{BSC},\ell}(g))$ is minimized when

$$
\begin{aligned}
\beta &= -\frac{\mathbb{E}_{\mathcal{D}_{\mathrm{c}},\mathcal{D}_{\mathrm{u}}}[AB]}{\mathbb{E}_{\mathcal{D}_{\mathrm{c}},\mathcal{D}_{\mathrm{u}}}[B^2]} - \frac{\mathbb{E}_{\mathcal{D}_{\mathrm{c}},\mathcal{D}_{\mathrm{u}}}[BC]}{\mathbb{E}_{\mathcal{D}_{\mathrm{c}},\mathcal{D}_{\mathrm{u}}}[B^2]}\alpha, \\
\alpha &= \frac{\mathbb{E}_{\mathcal{D}_{\mathrm{c}},\mathcal{D}_{\mathrm{u}}}[AB]\mathbb{E}_{\mathcal{D}_{\mathrm{c}},\mathcal{D}_{\mathrm{u}}}[BC] - \mathbb{E}_{\mathcal{D}_{\mathrm{c}},\mathcal{D}_{\mathrm{u}}}[AC]\mathbb{E}_{\mathcal{D}_{\mathrm{c}},\mathcal{D}_{\mathrm{u}}}[B^2]}{\mathbb{E}_{\mathcal{D}_{\mathrm{c}},\mathcal{D}_{\mathrm{u}}}[B^2]\mathbb{E}_{\mathcal{D}_{\mathrm{c}},\mathcal{D}_{\mathrm{u}}}[C^2] - \mathbb{E}^2_{\mathcal{D}_{\mathrm{c}},\mathcal{D}_{\mathrm{u}}}[BC]}.
\end{aligned}
$$

Substitute $\alpha$, we obtain

$$
\beta = \frac{\mathbb{E}_{\mathcal{D}_{\mathrm{c}},\mathcal{D}_{\mathrm{u}}}[AC]\mathbb{E}_{\mathcal{D}_{\mathrm{c}},\mathcal{D}_{\mathrm{u}}}[BC] - \mathbb{E}_{\mathcal{D}_{\mathrm{c}},\mathcal{D}_{\mathrm{u}}}[AB]\mathbb{E}_{\mathcal{D}_{\mathrm{c}},\mathcal{D}_{\mathrm{u}}}[C^2]}{\mathbb{E}_{\mathcal{D}_{\mathrm{c}},\mathcal{D}_{\mathrm{u}}}[B^2]\mathbb{E}_{\mathcal{D}_{\mathrm{c}},\mathcal{D}_{\mathrm{u}}}[C^2] - \mathbb{E}^2_{\mathcal{D}_{\mathrm{c}},\mathcal{D}_{\mathrm{u}}}[BC]}.
$$

Through plugging in the above formula, we have

$$
\begin{aligned}
\alpha &= \frac{n_{\mathrm{u}}}{n_{\mathrm{c}} + n_{\mathrm{u}}} - \frac{d_1 cov - cov^2}{d_1 d_2 - cov^2}\frac{n_{\mathrm{u}}}{n_{\mathrm{c}} + n_{\mathrm{u}}} + \frac{d_1 \sigma_{\mathrm{cov2}} - cov\sigma_{\mathrm{cov1}}}{d_1 d_2 - cov^2}\frac{n_{\mathrm{c}} n_{\mathrm{u}}}{n_{\mathrm{c}} + n_{\mathrm{u}}}, \\
\beta &= \frac{n_{\mathrm{u}}}{n_{\mathrm{c}} + n_{\mathrm{u}}} - \frac{d_2 cov - cov^2}{d_1 d_2 - cov^2}\frac{n_{\mathrm{u}}}{n_{\mathrm{c}} + n_{\mathrm{u}}} + \frac{d_2 \sigma_{\mathrm{cov1}} - cov\sigma_{\mathrm{cov2}}}{d_1 d_2 - cov^2}\frac{n_{\mathrm{c}} n_{\mathrm{u}}}{n_{\mathrm{c}} + n_{\mathrm{u}}}.
\end{aligned}
$$

$\square$

### B.4 PROOF OF THEOREM 4.3

**Theorem.** *Given a fixed agent policy $\pi_\theta$, the optimal discriminator $D_w^*(x)$ of Eq. (11) can be written as*

$$D_w^*(x) = \frac{(1-r)p + p_\theta}{p + p_\theta}.$$

*As a result, when the optimal discriminator $D_w^*(x)$ is given, the optimization of $\pi_\theta$ is equivalent to minimizing*

$$2\text{JSD}(p_\theta||p) - \text{KL}(p_\theta||p_1) - (1-\eta)\text{KL}(p_{\text{non}}||p_1) + C,$$

*where* $p_1 = (p_\theta + (1-\eta)p_{\text{non}})/(2-\eta)$, $C = \eta\mathbb{E}_{x\sim p_{\text{opt}}}\left[\log\frac{\eta p_{\text{opt}}}{p}\right] + (1-\eta)\mathbb{E}_{x\sim p_{\text{non}}}\left[\log\frac{(1-\eta)p_{\text{non}}}{p}\right] + \log(2-\eta) - (1-\eta)\log(1-\eta)/(2-\eta) - 2\log 2$, *which is a constant for $\pi_\theta$.*

*Proof.* Denote
$V(\pi_\theta, D_w) = \mathbb{E}_{x\sim p_\theta}\left[\log D_w(x)\right] + \mathbb{E}_{x\sim p}\left[r(x)\log(1 - D_w(x))\right] + \mathbb{E}_{x\sim p}\left[(1 - r(x))\log D_w(x)\right].$
We denote $D_w^*(x) = \arg\min_D V(\pi_\theta, D_w)$ and have

$$\frac{\partial V(\pi_\theta, D_w)}{\partial D} = \frac{p_\theta}{D} - \frac{rp}{1-D} + \frac{(1-r)p}{D}.$$

The maximum value of $V(\pi_\theta, D_w)$ occurs when its partial derivative with respect to $D$ is zero, given by $D_w = \frac{(1-r)p + p_\theta}{p + p_\theta}$. Consequently, we obtain $D_w^*(x) = \frac{(1-r)p + p_\theta}{p + p_\theta}$.

When the optimal discriminator $D_w^*(x)$ is given, $V(\pi_\theta, D_w)$ can be rewritten as

$$V(\pi_\theta, D_w) = \mathbb{E}_{x\sim p_\theta}\left[\log\frac{(1-r)p + p_\theta}{p + p_\theta}\right] + \mathbb{E}_{x\sim p}\left[r\log\frac{rp}{p + p_\theta}\right] + \mathbb{E}_{x\sim p}\left[(1-r)\log\frac{(1-r)p + p_\theta}{p + p_\theta}\right].$$

According to Eq. (9) and denoting $p_1 = (p_\theta + (1-\eta)p_{\text{non}})/(2-\eta)$, we obtain

$$V(\pi_\theta, D_w) = \mathbb{E}_{x\sim p_\theta}\left[\log\frac{(1-\eta)p_{\text{non}} + p_\theta}{p + p_\theta}\right] + \mathbb{E}_{x\sim p_{\text{opt}}}\left[\eta\log\frac{\eta p_{\text{opt}}}{p + p_\theta}\right] + \mathbb{E}_{x\sim p_{\text{non}}}\left[(1-\eta)\log\frac{(1-\eta)p_{\text{non}} + p_\theta}{p + p_\theta}\right]$$

$$= \mathbb{E}_{x\sim p_\theta}\left[-\log\frac{p_\theta}{(1-\eta)p_{\text{non}} + p_\theta} + \log\frac{p_\theta}{p + p_\theta}\right] + \eta\mathbb{E}_{x\sim p_{\text{opt}}}\left[\log\frac{\eta p_{\text{opt}}}{p + p_\theta}\right]$$

$$+ (1-\eta)\mathbb{E}_{x\sim p_{\text{non}}}\left[-\log\frac{(1-\eta)p_{\text{non}}}{(1-\eta)p_{\text{non}} + p_\theta} + \log\frac{(1-\eta)p_{\text{non}}}{p + p_\theta}\right]$$

$$= \mathbb{E}_{x\sim p_\theta}\left[\log\frac{p_\theta}{p + p_\theta}\right] + \eta\mathbb{E}_{x\sim p_{\text{opt}}}\left[\log\frac{\eta p_{\text{opt}}}{p + p_\theta}\right] + (1-\eta)\mathbb{E}_{x\sim p_{\text{non}}}\left[\log\frac{(1-\eta)p_{\text{non}}}{p + p_\theta}\right]$$

$$- \left(\mathbb{E}_{x\sim p_\theta}\left[\log\frac{p_\theta}{p_1}\right] - \underbrace{\log(2-\eta)}_{\text{const.w.r.t.}\pi_\theta} + (1-\eta)\mathbb{E}_{x\sim p_{\text{non}}}\left[\log\frac{p_{\text{non}}}{p_1}\right] + \underbrace{(1-\eta)\log\frac{1-\eta}{2-\eta}}_{\text{const.w.r.t.}\pi_\theta}\right)$$

$$= \mathbb{E}_{x\sim p_\theta}\left[\log\frac{p_\theta}{p + p_\theta}\right] + \eta\mathbb{E}_{x\sim p_{\text{opt}}}\left[\log\frac{\eta p_{\text{opt}}}{p} + \log\frac{p}{p + p_\theta}\right] + (1-\eta)\mathbb{E}_{x\sim p_{\text{non}}}\left[\log\frac{(1-\eta)p_{\text{non}}}{p}\right.$$

$$\left. + \log\frac{p}{p + p_\theta}\right] - \text{KL}(p_\theta||p_1) - (1-\eta)\text{KL}(p_{\text{non}}||p_1) + C_1$$

$$= \mathbb{E}_{x\sim p_\theta}\left[\log\frac{p_\theta}{p + p_\theta}\right] + \mathbb{E}_{x\sim p}\left[\log\frac{p}{p + p_\theta}\right] + \underbrace{\eta\mathbb{E}_{x\sim p_{\text{opt}}}\left[\log\frac{\eta p_{\text{opt}}}{p}\right] + (1-\eta)\mathbb{E}_{x\sim p_{\text{non}}}\left[\log\frac{(1-\eta)p_{\text{non}}}{p}\right]}_{\text{const.w.r.t.}\pi_\theta}$$

$$- \text{KL}(p_\theta||p_1) - (1-\eta)\text{KL}(p_{\text{non}}||p_1) + C_1$$

$$= \mathbb{E}_{x\sim p_\theta}\left[\log\frac{p_\theta}{(p + p_\theta)/2}\right] + \mathbb{E}_{x\sim p}\left[\log\frac{p}{(p + p_\theta)/2}\right] - 2\log 2 - \text{KL}(p_\theta||p_1) - (1-\eta)\text{KL}(p_{\text{non}}||p_1) + C_2$$

$$= 2\text{JSD}(p_\theta||p) - \text{KL}(p_\theta||p_1) - (1-\eta)\text{KL}(p_{\text{non}}||p_1) + C,$$

where $C = \eta\mathbb{E}_{x\sim p_{\text{opt}}}\left[\log\frac{\eta p_{\text{opt}}}{p}\right] + (1-\eta)\mathbb{E}_{x\sim p_{\text{non}}}\left[\log\frac{(1-\eta)p_{\text{non}}}{p}\right] + \log(2-\eta) - (1-\eta)\log(1-\eta)/(2-\eta) - 2\log 2$ is a constant for $\pi_\theta$. $\qquad\square$

## B.5 Proof of Theorem 4.4

**Theorem.** *Denote $\mathcal{G}$ as the hypothesis class being utilized and $\mathfrak{R}_n(\mathcal{G})$ as the Rademacher complexity of the function class $\mathcal{G}$ with a sample size of $n$. Assume that the loss function $\ell$ is $\rho_\ell$-Lipschitz continuous, and there exists a constant $C_\ell > 0$ such that for any $g \in \mathcal{G}$, $\sup_{x\in\mathcal{X}, y\in\{\pm 1\}}|\ell(yg(x))| \leq C_\ell$. Define $\hat{g}$ as the minimizer of $\widehat{R}_{\text{BSC},\ell}(g)$ over $g \in \mathcal{G}$ and $g^*$ as the minimizer of $R_{\text{BSC},\ell}(g)$ over $g \in \mathcal{G}$. For $\delta \in (0,1)$, with probability at least $1 - \delta$ when repeatedly sampling data to train $\hat{g}$, we have*

$$
\begin{aligned}
R_{\text{BSC},\ell}(\hat{g}) - R_{\text{BSC},\ell}(g^*) \leq &16\rho_L((3+\alpha-\beta)\mathfrak{R}_{n_{\text{c}}}(\mathcal{G}) + (\alpha+\beta)\mathfrak{R}_{n_{\text{u}}}(\mathcal{G})) \\
&+ 4C_L\sqrt{\frac{\log(12/\delta)}{2}}\left((3+\alpha-\beta)n_{\text{c}}^{-\frac{1}{2}} + (\alpha+\beta)n_{\text{u}}^{-\frac{1}{2}}\right).
\end{aligned}
$$

*Proof.* Like 2IWIL, since $\hat{g}$ and $g^*$ are the minimizers of $\widehat{R}_{\text{BSC},\ell}(g)$ and $R_{\text{BSC},\ell}(g)$, respectively, we have

$$
\begin{aligned}
R_{\text{BSC},\ell}(\hat{g}) - R_{\text{BSC},\ell}(g^*) &= R_{\text{BSC},\ell}(\hat{g}) - \widehat{R}_{\text{BSC},\ell}(\hat{g}) + \widehat{R}_{\text{BSC},\ell}(\hat{g}) - \widehat{R}_{\text{BSC},\ell}(g^*) + \widehat{R}_{\text{BSC},\ell}(g^*) - R_{\text{BSC},\ell}(g^*) \\
&\leq \sup_{g\in\mathcal{G}}\left(R_{\text{BSC},\ell}(g) - \widehat{R}_{\text{BSC},\ell}(g)\right) + 0 + \sup_{g\in\mathcal{G}}\left(\widehat{R}_{\text{BSC},\ell}(g) - R_{\text{BSC},\ell}(g)\right) \\
&\leq 2\sup_{g\in\mathcal{G}}\left|\widehat{R}_{\text{BSC},\ell}(g) - R_{\text{BSC},\ell}(g)\right|.
\end{aligned}
$$

Hence, we just need to bound the uniform deviation $\sup_{g\in\mathcal{G}}\left|\widehat{R}_{\text{BSC},\ell}(g) - R_{\text{BSC},\ell}(g)\right|$. We have

$$
\begin{aligned}
&\sup_{g\in\mathcal{G}}\left|\widehat{R}_{\text{BSC},\ell}(g) - R_{\text{BSC},\ell}(g)\right| \\
&\leq \sup_{g\in\mathcal{G}}\left|\frac{1}{n_{\text{c}}}\sum_{i=1}^{n_{\text{c}}}(r_i(\ell(g(x_{\text{c},i})) - \ell(-g(x_{\text{c},i}))) + (1-\beta)\ell(-g(x_{\text{c},i})) - \alpha\ell(g(x_{\text{c},i})))\right. \\
&\qquad \left. - \mathbb{E}_{x,r\sim q}\left[r(\ell(g(x)) - \ell(-g(x))) + (1-\beta)\ell(-g(x)) - \alpha\ell(g(x))\right]\right| \\
&\quad + \beta\sup_{g\in\mathcal{G}}\left|\frac{1}{n_{\text{u}}}\sum_{i=1}^{n_{\text{u}}}\ell(-g(x_{\text{u},i})) - \mathbb{E}_{x\sim p}\left[\ell(-g(x))\right]\right| + \alpha\sup_{g\in\mathcal{G}}\left|\frac{1}{n_{\text{u}}}\sum_{i=1}^{n_{\text{u}}}\ell(g(x_{\text{u},i})) - \mathbb{E}_{x\sim p}\left[\ell(g(x))\right]\right| \\
&\leq \sup_{g\in\mathcal{G}}\left|\frac{1}{n_{\text{c}}}\sum_{i=1}^{n_{\text{c}}}r_i\ell(g(x_{\text{c},i})) - \mathbb{E}_{x,r\sim q}[r\ell(g(x))]\right| + \sup_{g\in\mathcal{G}}\left|\frac{1}{n_{\text{c}}}\sum_{i=1}^{n_{\text{c}}}r_i\ell(-g(x_{\text{c},i})) - \mathbb{E}_{x,r\sim q}[r\ell(-g(x))]\right| \\
&\quad + (1-\beta)\sup_{g\in\mathcal{G}}\left|\frac{1}{n_{\text{c}}}\sum_{i=1}^{n_{\text{c}}}\ell(-g(x_{\text{c},i})) - \mathbb{E}_{x,r\sim q}[\ell(-g(x))]\right| + \beta\sup_{g\in\mathcal{G}}\left|\frac{1}{n_{\text{u}}}\sum_{i=1}^{n_{\text{u}}}\ell(-g(x_{\text{u},i})) - \mathbb{E}_{x\sim p}[\ell(-g(x))]\right| \\
&\quad + \alpha\sup_{g\in\mathcal{G}}\left|\frac{1}{n_{\text{c}}}\sum_{i=1}^{n_{\text{c}}}\ell(g(x_{\text{c},i})) - \mathbb{E}_{x,r\sim q}[\ell(g(x))]\right| + \alpha\sup_{g\in\mathcal{G}}\left|\frac{1}{n_{\text{u}}}\sum_{i=1}^{n_{\text{u}}}\ell(g(x_{\text{u},i})) - \mathbb{E}_{x\sim p}[\ell(g(x))]\right|.
\end{aligned}
$$

According to 2IWIL (Theorem 4.3 in Wu et al. (2019)), since the above six terms are the bounded differences with constants $C_L/n_{\text{c}}$, $C_L/n_{\text{c}}$, $C_L/n_{\text{c}}$, $C_L/n_{\text{u}}$, $C_L/n_{\text{c}}$, and $C_L/n_{\text{u}}$, respectively, we

can bound them with probability at least $1 - \delta/6$ as

$$\sup_{g \in \mathcal{G}} \left| \frac{1}{n_c} \sum_{i=1}^{n_c} r_i \ell(g(x_{c,i})) - \mathbb{E}_{x,r \sim q}[r\ell(g(x))] \right| \leq 8\rho_L \mathfrak{R}_{n_c}(\mathcal{G}) + 2C_L \sqrt{\frac{\log(12/\delta)}{2n_c}},$$

$$\sup_{g \in \mathcal{G}} \left| \frac{1}{n_c} \sum_{i=1}^{n_c} r_i \ell(-g(x_{c,i})) - \mathbb{E}_{x,r \sim q}[r\ell(-g(x))] \right| \leq 8\rho_L \mathfrak{R}_{n_c}(\mathcal{G}) + 2C_L \sqrt{\frac{\log(12/\delta)}{2n_c}},$$

$$\sup_{g \in \mathcal{G}} \left| \frac{1}{n_c} \sum_{i=1}^{n_c} \ell(-g(x_{c,i})) - \mathbb{E}_{x,r \sim q}[\ell(-g(x))] \right| \leq 8\rho_L \mathfrak{R}_{n_c}(\mathcal{G}) + 2C_L \sqrt{\frac{\log(12/\delta)}{2n_c}},$$

$$\sup_{g \in \mathcal{G}} \left| \frac{1}{n_u} \sum_{i=1}^{n_u} \ell(-g(x_{u,i})) - \mathbb{E}_{x \sim p}[\ell(-g(x))] \right| \leq 8\rho_L \mathfrak{R}_{n_u}(\mathcal{G}) + 2C_L \sqrt{\frac{\log(12/\delta)}{2n_u}},$$

$$\sup_{g \in \mathcal{G}} \left| \frac{1}{n_c} \sum_{i=1}^{n_c} \ell(g(x_{c,i})) - \mathbb{E}_{x,r \sim q}[\ell(g(x))] \right| \leq 8\rho_L \mathfrak{R}_{n_c}(\mathcal{G}) + 2C_L \sqrt{\frac{\log(12/\delta)}{2n_c}},$$

$$\sup_{g \in \mathcal{G}} \left| \frac{1}{n_u} \sum_{i=1}^{n_u} \ell(g(x_{u,i})) - \mathbb{E}_{x \sim p}[\ell(g(x))] \right| \leq 8\rho_L \mathfrak{R}_{n_u}(\mathcal{G}) + 2C_L \sqrt{\frac{\log(12/\delta)}{2n_u}}.$$

In the end, we can bound the initial estimation error with probability of at least $1 - \delta$:

$$R_{\mathrm{BSC},\ell}(\hat{g}) - R_{\mathrm{BSC},\ell}(g^*) \leq 16\rho_L((3 + \alpha - \beta)\mathfrak{R}_{n_c}(\mathcal{G}) + (\alpha + \beta)\mathfrak{R}_{n_u}(\mathcal{G}))$$
$$+ 4C_L \sqrt{\frac{\log(12/\delta)}{2}} \left( (3 + \alpha - \beta)n_c^{-\frac{1}{2}} + (\alpha + \beta)n_u^{-\frac{1}{2}} \right).$$

$\square$

## C   DETAILS OF THE EXPERIMENTS AND ADDITIONAL EXPERIMENTS

### C.1   HYPER-PARAMETERS SETTINGS AND TASK INFORMATION

All of our experiments are run on a single machine with 4 NVIDIA GeForce RTX 3080 GPUs. For the architectures of all neural networks, we employ two hidden layers of size 100 each, using Tanh as the activation function. Across all tasks, we utilize the same hyper-parameters as listed in Table 2. Table 3 shows the number of confidence data and unlabeled data used for each task, along with the cumulative rewards corresponding to the optimal and random policies.

Table 2: Hyper-parameters settings

| Hyper-parameters | value |
| --- | --- |
| $\gamma$ | 0.995 |
| $\tau$ (Generalized Advantage Estimation) | 0.97 |
| Batch size | $5,000$ |
| Learning rate (Value network) | $3 \times 10^{-4}$ |
| Learning rate (Discriminator) | $1 \times 10^{-3}$ |
| Learning rate (Classifier) | $3 \times 10^{-4}$ |
| Optimizer | Adam |

### C.2   CLIP FUNCTION IN BSC

In the process of deriving Eq. (13), two terms with theoretical values of $0$ are introduced: $\mathbb{E}_{x \sim p}[\alpha\ell(g(x))] - \mathbb{E}_{x \sim q}[\alpha\ell(g(x))]$ and $\mathbb{E}_{x \sim p}[\beta\ell(-g(x))] - \mathbb{E}_{x \sim q}[\beta\ell(-g(x))]$. If we directly minimize Eq. (13), the two terms may deviate far from $0$. To mitigate this issue, we use the clip function to limit the sum of these two items to a neighborhood of $0$. Specifically, we denote

Table 3: Task information

| Tasks | $n_{\mathrm{c}}$ | $n_{\mathrm{u}}$ | Optimal policy | Random policy |
|---|---|---|---|---|
| Ant-v2 | 120 | 480 | 4143.10 | -72.30 |
| HalfCheetah-v2 | 500 | 2000 | 3467.32 | -288.44 |
| Hopper-v2 | 20 | 80 | 3250.67 | 18.04 |
| Pendulum-v1 | 200 | 800 | -116.81 | -1200.96 |
| Swimmer-v2 | 5 | 20 | 348.99 | 2.31 |
| Walker-v2 | 400 | 1600 | 3694.13 | 1.91 |

$R_1 = \mathbb{E}_{x \sim p}[\alpha \ell(g(x))] - \mathbb{E}_{x \sim q}[\alpha \ell(g(x))] + \mathbb{E}_{x \sim p}[\beta \ell(-g(x))] - \mathbb{E}_{x \sim q}[\beta \ell(-g(x))]$. According to Eq. (13), the final empirical risk can be written as

$$\widehat{R}_{\mathrm{BSC},\ell}(g) = \widehat{R}_C(g) + \mathrm{clip}_{[-\epsilon,\epsilon]}\widehat{R}_1,$$

where

$$\widehat{R}_C(g) = \frac{1}{n_{\mathrm{c}}} \sum_{i=1}^{n_{\mathrm{c}}} \left[ r_i \ell(g(x_{\mathrm{c},i})) + (1 - r_i)\ell(-g(x_{\mathrm{c},i})) \right].$$

In Walker2d-v2, the epoch for classifier training is set to $50000$, with $\epsilon$ configured to $0.05$. In other experiments, the epoch for classifier training is $25000$, with $\epsilon$ configured to $0.01$.

## C.3 ADDITIONAL EXPERIMENTS

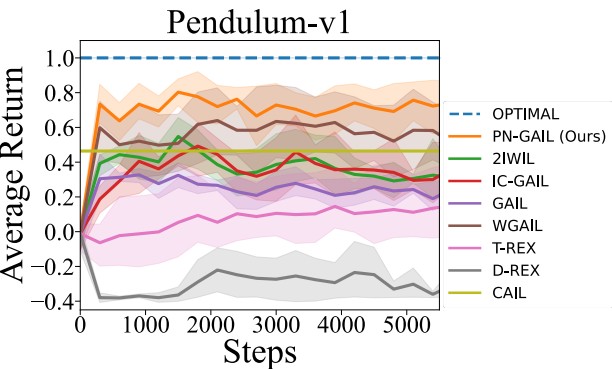

Figure 5: Experiments with baselines including weighting-based and ranking-based methods.

CAIL uses the average reward after training convergence, and the implementation is based on CAIL codebase. As shown in Fig. 5, PN-GAIL outperforms other baseline methods, achieving the highest return.

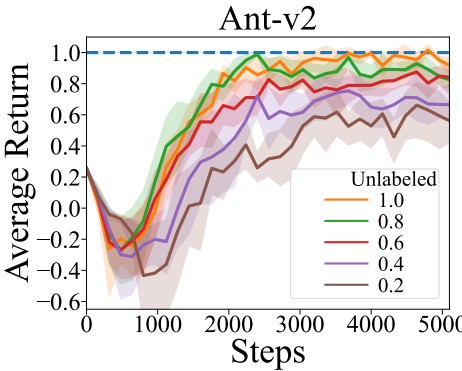

Figure 6: Experiments on reducing the number of unlabeled confidence demonstrations.

We gradually reduce the proportion of unlabeled confidence demonstrations. In Fig. 6, the numbers in the legend indicate the proportion of unlabeled data used as demonstrations. The performance of PN-GAIL improves as the amount of unlabeled data increases, illustrating how the use of unlabeled data can enhance the performance of our method.

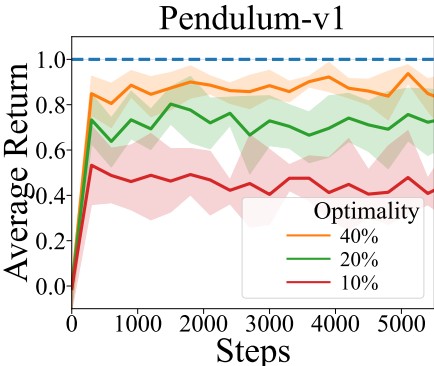

Figure 7: Experiments under different optimalities of imperfect demonstrations.

We evaluate the performance of PN-GAIL under different optimalities of imperfect demonstrations in the Pendulum-v1 environment. In Fig. 7, the numbers in the legend illustrate the proportion of demonstrations generated by the optimal policy when collecting datasets ($\pi_{\mathrm{opt}} : \pi_1 = 2 : 3$, $\pi_{\mathrm{opt}} : \pi_1 = 1 : 4$, $\pi_{\mathrm{opt}} : \pi_1 = 1 : 9$). The higher the proportion of demonstrations generated by the optimal policy is, the better the performance of PN-GAIL.

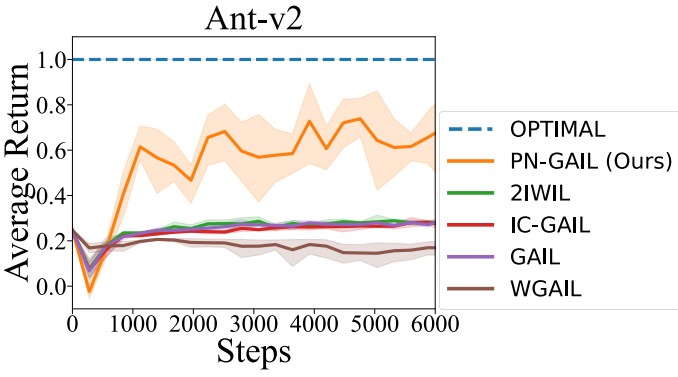

Figure 8: Experiments with an extreme demonstration ratio.

In the Ant-v2 environment, we train an optimal policy $\pi_{\text{opt}}$ and an intermediate policy $\pi_1$ using TRPO. We collect demonstrations with ratio of $\pi_{\text{opt}} : \pi_1 = 1 : 10$ and all demonstrations use normalized rewards to annotate confidence scores. We then evaluate the performance of PN-GAIL and other baseline methods using the collected demonstrations. As shown in Fig. 8, under this extreme demonstration ratio, all other methods fail, the final result is close to GAIL, and only PN-GAIL learns valid information from the extreme demonstrations.

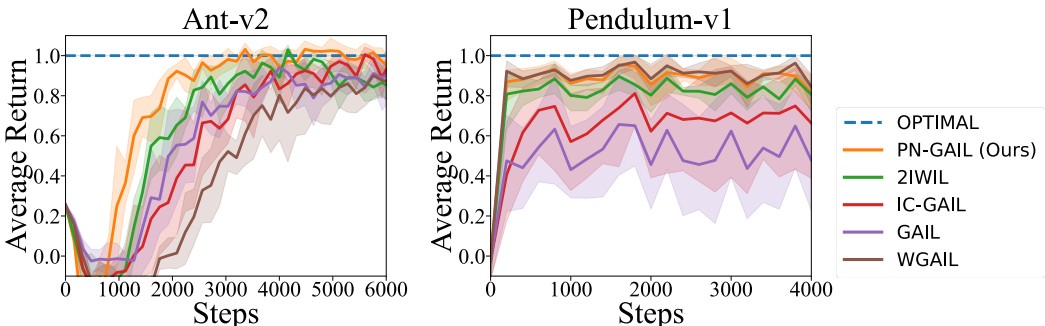

Figure 9: Experiments when optimal demonstrations are dominant.

In the Ant-v2 and Pendulum-v1 environments, we conduct the experiments at the demonstration ratio of $\pi_{\text{opt}} : \pi_1 = 2 : 1$. As shown in Fig. 9, it can be seen that when the optimal demonstrations are dominant, PN-GAIL still shows robust and excellent performance.

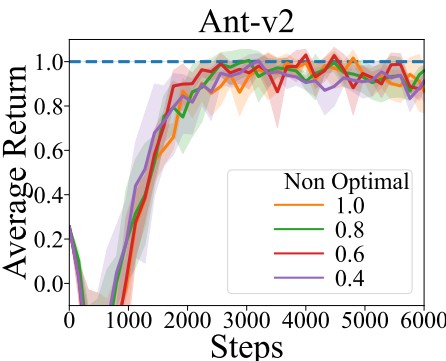

Figure 10: Experiments under different number of non-optimal demonstrations.

We conduct experiments with different numbers of non-optimal demonstrations in the Ant-v2 environment. In Fig. 10, the numbers in the legend indicate the proportion of non-optimal demonstrations used as demonstrations, and the blue dotted line represents the performance of the optimal policy. When the number of non-optimal demonstrations decreases, the performance of PN-GAIL does not decrease significantly. We speculate that PN-GAIL does not take advantage of additional non-optimal demonstrations now because the number of optimal demonstrations in given expert demonstrations is sufficient to learn an optimal policy.

Since the ranking-based methods including CAIL and f-IRL all require demonstrations to be stored in the form of a trajectory, we only evaluate them in the Pendulum-v1 and Ant-v2 environments. The dataset used in the Pendulum-v1 environment is the same as in Fig. 2, and the dataset used in the Ant-v2 environment is the same as in Fig. 8. The implementation of CAIL, T-REX and D-REX is based on the CAIL codebase, and the implementation of f-IRL is based on the f-IRL codebase. All methods use the average reward after training convergence. We construct trajectory rankings based on confidence scores, and the average returns (only for evaluation) across all methods are shown in Table 4. PN-GAIL outperforms other methods, achieving the highest returns.

Table 4: Average returns of PN-GAIL, CAIL, ranking-based methods and f-IRL during training (only for evaluation).

| Methods | Pendulum-v1 | Ant-v2 |
|---|---|---|
| PN-GAIL (Ours) | **-465.978 ± 132.710** | **2960.458 ± 851.465** |
| CAIL | -697.821 ± 91.459 | 2336.546 ± 735.378 |
| T-REX | -1074.435 ± 258.078 | -1858.025 ± 217.228 |
| D-REX | -1532.958 ± 114.138 | -2495.473 ± 220.703 |
| FKL(f-IRL) | -698.064 ± 300.793 | 1563.796 ± 1372.482 |
| RKL(f-IRL) | -603.760 ± 189.727 | 962.785 ± 816.547 |
| JS(f-IRL) | -581.602 ± 185.096 | 882.026 ± 690.371 |
| Optimal Policy | -116.81 | 4271.79 |

Table 5: The variance of the estimator $\hat{R}_{BSC,\ell}(g)$.

| Var | Ant-v2 | HalfCheetah-v2 | Hopper-v2 | Pendulum-v1 | Swimmer-v2 | Walker2d-v2 |
|---|---|---|---|---|---|---|
| Origin | 0.126±0.054 | 0.033±0.009 | 1.476±0.574 | 0.00064±0.00024 | 1.664±1.475 | 0.093±0.029 |
| PN-GAIL(Ours) | **0.100±0.050** | **0.025±0.006** | **1.412±0.566** | **0.00013±0.00004** | **0.007±0.015** | **0.084±0.028** |

Origin indicates the lack of $\alpha$ and $\beta$ (i.e., $\alpha$=0 and $\beta$=0). The results in Table 5 show that the variance of PN-GAIL is consistently smaller, demonstrating the validity of the chosen values for $\alpha$ and $\beta$.

## C.4 UNCROPPED FIGURES OF ANT-V2

In our experiments, we observe a decrease followed by an increase in performance within the Ant-v2 environment. For better comparison, we have cropped the figures of Ant-v2 in the main text, while the uncropped figures are presented below:

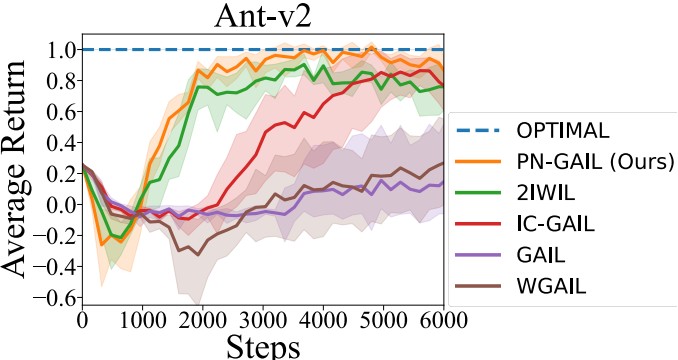

Figure 11: Normalized average returns of PN-GAIL and baseline methods during training.

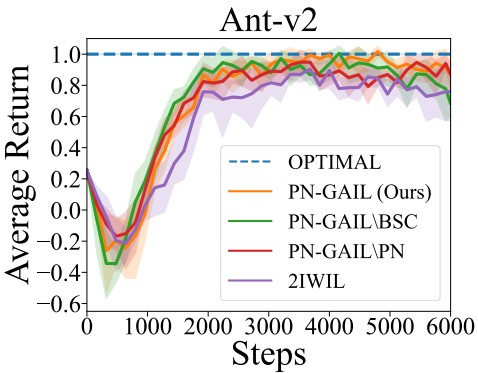

Figure 12: Normalized average returns of ablation experiments during training.

