# OpenReview forum: "PN-GAIL: Leveraging Non-optimal Information from Imperfect Demonstrations"
_ICLR.cc/2025/Conference — ICLR 2025 Poster_

### Official Review · Reviewer_1f3M · 2024-10-29

**Soundness:** 4
**Presentation:** 4
**Contribution:** 3
**Rating:** 8
**Confidence:** 3

**Summary:**

Motivated by the limitation of 2IWIL in a type of scenario where certain non-optimal demonstrations have high probabilities of appearing in a set of imperfect (unlabeled) demonstrations, the paper proposes a new method named PN-GAIL, better leveraging optimal and non-optimal information from imperfect demonstrations to learn optimal reward and policy by assessing both positive and negative risks. Moreover, the paper modifies semi-conf classification in 2IWIL to establish balanced semi-conf classification to handle better the cases where certain demonstrations only exist in the confidence (known, labeled) data.

The authors conduct experiments, comparing their PN-GAIL to four baseline methods across six environments. The results show that PN-GAIL alleviates the impact of the unbalanced frequency in impact demonstrations, outperforms other methods, and maintains relatively good performances given the decreasing number of labels. Also, the outcomes demonstrate that the balanced semi-conf classifier improves performances, particularly in three out of six environments.

**Strengths:**

1. The main objective/purpose of this paper is articulated explicitly with the existing methods and their limitations highlighted clearly.
2. The literature review on IL, GAIL, and IL with imperfect information is comprehensive. The preliminaries provide the essential information of 2IWIL.
3. The theoretical derivations regarding the discriminator modification and classification refinement are straightforward and concise in the main body, which makes audience easy to follow; meanwhile, the supplemental materials in the appendices provide necessary, detailed explanations.
4. The experiments with three goals setup are tightly related to the main achievements that the paper wants to claim. The experiments are conducted with representative benchmarks across various environments, effectively showing the performances with well-formatted figures and table in the main context.
5. Overall, the authors identify the potential challenges of 2IWIL about certain non-optimal data with high frequencies in an unlabeled demonstration set significantly affecting reward and policy generation. The topic is likely to be of interest to a large proportion of the community. The proposed PN-GAIL creatively update the previous methods to successfully remove limitations of prior IL results to a certain extent.

**Weaknesses:**

1. Missing definition: in Section 3, what δ represents. Please define δ when it is first introduced in the paper.

2. Needing clarification: PN-GAIL\BSC: PN-GAIL without balanced semi-conf (BSC) classification--does this mean no classification used or SC used? Without a probabilistic classifier, how to obtain confidence scores? Please explicitly state whether a SC classification is used, and to explain how confidence scores are obtained if no classifier is used.

3. Lack of analysis for experimental outcomes: it is necessary to provide more detailed explanations and discussions regarding those figures. For example, in Figure 3 what possible reasons (e.g., characteristics of each environment? limited number of demonstrations?) are that result in the varying performance patterns across six environments. We can observe that in some cases three colors (methods) are similar; in some cases blue and green are close; in some cases, blue and orange are close; while in some cases, blue works best. Please provide a systematic analysis of how different factors might contributing to these patterns.

**Questions:**

1. About clarification: PN-GAIL\BSC: PN-GAIL without balanced semi-conf (BSC) classification--does this mean no classification used or SC used? Without a probabilistic classifier, how to obtain confidence scores?

2. To verify BSC outperform SC in 2IWIL, straightforward comparisons are PN-GAIL(BSC) vs. PN-GAIL(switch to SC) vs. 2IWIL(switch to BSC) vs. 2IWIL(SC). Why do you consider the way of comparisons that are presented in the paper?

3. It is better to provide more detailed explanations and discussions regarding those figures. For example, the statement about Figure 3 is quite short. The audience would like to learn about what possible reasons are that result in the varying performance patterns across six environments in Figure 3. We can observe that in some cases three colors (methods) are similar; in some cases blue and green are close; in some cases, blue and orange are close; while in some cases, blue works best. Could you provide a systematic analysis?

---

> ### Author Response · Authors · 2024-11-20
> **Author Response to Reviewer 1f3M**
>
> > **Q1.** Missing definition: in Section 3, what $\delta$ represents. Please define $\delta$ when it is first introduced in the paper.
>
> **A1.** Thank you for pointing it out! $\delta$ represents Dirac delta function. We have added the above explanation to the revised version.
>
> > **Q2.** Needing clarification: PN-GAIL$\backslash$BSC: PN-GAIL without balanced semi-conf (BSC) classification--does this mean no classification used or SC used?
>
> **A2.** We apologize for the inaccurate statement. PN-GAIL$\backslash$BSC stands for using SC classification. We have clarified the relevant explanations in the revised version.
>
>
> > **Q3.** Lack of analysis for experimental outcomes: it is necessary to provide more detailed explanations and discussions regarding those figures.
>
> **A3.** Thank you for your suggestion. In Figure 3, the difference between the performance of PN-GAIL and PN-GAIL$\backslash$PN indicates that there is a preference in the imperfect demonstrations, resulting in the poor performance of the 2IWIL follow-up method. The performance gap between the performance of PN-GAIL and PN-GAIL$\backslash$BSC indicates that the prediction confidence of SC classification is not accurate enough, which affects the subsequent training. If the performance gap is not significant, it means that the above problems are not obvious or do not affect the final results. For example, in the Pendulum-v1 environment, we designed a high percentage of non-optimal demonstrations. In Figure 3, the performance of PN-GAIL and PN-GAIL$\backslash$BSC are close to and higher than that of PN-GAIL$\backslash$PN, indicating that there is little difference between the BSC classification and the SC classification, and there is a preference in imperfect demonstrations. This is also in line with the design of our dataset and the results of the confidence classifier. We have added the above explanation to the revised version.
>
>
> > **Q4.** To verify BSC outperform SC in 2IWIL, straightforward comparisons are PN-GAIL(BSC) vs. PN-GAIL(switch to SC) vs. 2IWIL(switch to BSC) vs. 2IWIL(SC). Why do you consider the way of comparisons that are presented in the paper?
>
> **A4.** This is because a direct comparison of the difference between the classification results and the true value is more accurate, convincing, and quantifiable. We have compared the performance of PN-GAIL(BSC) with PN-GAIL$\backslash$BSC (which means using SC) in Figure 3. In addition, we have also added comparison between PN-GAIL$\backslash$PN (2IWIL with BSC) and 2IWIL (SC) in Figure 3. The results of both comparisons suggest that BSC is superior to SC.

---

> > ### Comment · Reviewer_1f3M · 2024-11-20
> >
> > Thank you for your response to my questions. I appreciate the incremental experiments you conducted and the modifications you made. I confirm rating score 8 as the authors did great work regarding initial submission and revision.

---

> > > ### Author Response · Authors · 2024-11-21
> > > **Thanks for your response!**
> > >
> > > Thank you so much for taking the time and effort to review our paper and read the follow-up rebuttal! We are pleased to hear that you recognize our work and would like to express our sincere appreciation for your feedback.

---

### Official Review · Reviewer_XEhc · 2024-10-31

**Soundness:** 3
**Presentation:** 3
**Contribution:** 3
**Rating:** 6
**Confidence:** 4

**Summary:**

The authors propose PN-GAIL, an extension of the GAIL framework designed to handle the imperfect expert demonstrations. By predicting confidence scores for unlabeled data, PN-GAIL allows for more accurate imitation learning without relying solely on optimal examples. Theoretical analysis is provided for the output of the optimal discriminator in the proposed method.

**Strengths:**

This paper introduces a new algorithm (based on 2IWIL and IC-GAIL) supported by theoretical analysis and demonstrates its effectiveness across multiple tasks. Experiments are extensive, including different benchmarks, different $\pi_{OPT} : \pi_{1}$ ratios, different standard deviations of Gaussian noise.

**Weaknesses:**

Although the proposed method is based on [1], e.g., some techniques to prove Theorem 4.1, Theorem 4.2, have been explored in [1], this study sill broadens the scope of [1]. One potential weakness is: although $\alpha$ and $\beta$ are intended to play distinct roles in Theorem 4.2, they are selected to be identical in Algorithm 1, which may affect the overall applicability of the algorithm.

[1] Wu, Yueh-Hua, et al. "Imitation learning from imperfect demonstration." International Conference on Machine Learning. PMLR, 2019.

**Questions:**

1. Could the authors highlight expert performance more prominently in the figures to enhance clarity and interpretability?

2. Would varying values of $n_{u}$ and $n_{c}$ significantly impact the performance of the proposed method? Additionally, could the authors provide guidelines or criteria for selecting optimal values for these parameters in practice?

3. Figure 1 offers a valuable comparison of the proposed method against GAIL and 2IWIL. To further support this comparison, it would be better if the authors could also provide more intuition of why GAIL and 2IWIL fail but the proposed method succeed. E.g., gradient colors for different confidence scores, and why 2IWIL fails to predict them. Additionally, Why GAIL predicts 5.0 for some of theses data points?

4. To further contextualize this work within imitation learning (specifically, driver behavior imitation), it would be beneficial to incorporate additional relevant studies, such as:

[2] Ruan, Kangrui, and Xuan Di. "Learning human driving behaviors with sequential causal imitation learning." Proceedings of the AAAI Conference on Artificial Intelligence. Vol. 36. No. 4. 2022.

[3] Hawke, Jeffrey, et al. "Urban driving with conditional imitation learning." 2020 IEEE International Conference on Robotics and Automation (ICRA). IEEE, 2020.

and so on.

---

> ### Author Response · Authors · 2024-11-20
> **Author Response to Reviewer XEhc**
>
> > **Q1.** Although $\alpha$ and $\beta$ are intended to play distinct roles in Theorem 4.2, they are selected to be identical in Algorithm 1, which may affect the overall applicability of the algorithm.
>
> **A1.** We do not directly choose $\alpha$ and $\beta$ as the same value. In fact, after assuming that the covariances are small enough relative to the variances, the final approximations of $\alpha$ and $\beta$ happen to be the same. In addition, we empirically validate the appropriateness of our chosen values of $\alpha$ and $\beta$ in **Common Response point 6**. The results demonstrate the rationality of this setting.
>
>
> > **Q2.** Could the authors highlight expert performance more prominently in the figures to enhance clarity and interpretability?
>
> **A2.** Thank you for your valuable suggestion. Following your suggestion, we have redrawn Figures 2, 3, 4 and added a blue dotted line to indicate the performance of the optimal policy.
>
>
> > **Q3.** Would varying values of $n_u$ and $n_c$ significantly impact the performance of the proposed method?
>
> **A3.** We show the PN-GAIL performance at different $n_u$ and $n_c$ ratios in Figure 4 (a), Figure 4(b), and the PN-GAIL performance when $n_u$ is reduced in **Appendix B.3 Figure 6**. The results show that decreasing $n_u$ has a greater impact on method performance than changing the ratio of $n_u$ and $n_c$. In practice, when choosing the dataset size ($n_u+n_c$), it is better to ensure that the optimal demonstrations information contained in the dataset is sufficient to learn an optimal policy. When selecting the label ratio ($\frac{n_c}{n_c + n_u}$), we need to make the distribution of the label dataset as consistent as possible with the distribution of the whole dataset.
>
>
> > **Q4.** Provide more intuition of why GAIL and 2IWIL fail, but the proposed method succeeds. Why does GAIL predict 5.0 for some of these data points?
>
> **A4.** Thank you for your valuable comments. We have redrawn Figure 1 and used gradient colors for different confidence scores. We explain Figure 1 with an example in Section 4.1, where the data points represent the equivalent confidence level at the time of training. Specifically, we consider the goals of GAIL, 2IWIL:
>
> $\min_\theta \max_w E_{x\sim p_\theta}\left[\log D_w(x)\right]+ E_{x\sim p}\left[\frac{r(x)}{\eta}\log(1-D_w(x))\right].$
>
> Expand the second term, which is $\sum p(x) \frac{r(x)}{\eta}\log(1-D_w(x))$, and since GAIL has no confidence scores, $\frac{r(x)}{\eta}$ is always equal to $1$. The coefficient for each state action pair $x$ in given demonstrations at the time of training is $p(x) \frac{r(x)}{\eta}$. Therefore, if there is a higher probability of $x_1$ appearing in imperfect demonstrations, such as $p(x_1) = 5p(x_{other})$. Then its coefficient is $p(x_1)\frac{r(x_1)}{\eta} = p(x_{other})\frac{5r(x_1)}{\eta}$, which equates to the confidence score of $x_1$, five times that of the original. For GAIL, since the confidence scores are all considered to be $1.0$, it is $1.0\times5=5.0$.
>
> For PN-GAIL, by increasing the non-optimal information of the demonstrations, the higher the probability of a demonstration, the third term of Eq 11 will balance the influence of the second item due to the weight change, so that the score given by discriminator $D$ is more accurate. We have added the above explanations to the revised version.
>
>
> > **Q5.** To further contextualize this work within imitation learning (specifically, driver behavior imitation), it would be beneficial to incorporate additional relevant studies.
>
> **A5.** In fact, we have already attempted the proposed algorithm in the field of autonomous driving and achieved preliminary results. Due to time and space limitations, we will publish such results in our future work.

---

> ### Comment · Reviewer_XEhc · 2024-11-22
> **Official Comment by Reviewer XEhc**
>
> Thanks to the authors for providing a detailed response. Based on the revisions and clarifications, I believe this paper meets the standards for acceptance. As such, I decide to maintain my score.
>
> For the final camera-ready version, I encourage the authors to consider citing additional relevant references (e.g., the one we reviewers suggested [2][3][4][5] and some recently published works) to further strengthen the paper’s impact and completeness. While I totally understand that conducting additional experiments may be challenging at this stage, it would still be valuable to include comparisons and discussions in the related work section.
>
> driver behavior imitation:
>
> [2] Ruan, Kangrui, and Xuan Di. "Learning human driving behaviors with sequential causal imitation learning." Proceedings of the AAAI Conference on Artificial Intelligence. Vol. 36. No. 4. 2022.
>
> [3] Hawke, Jeffrey, et al. "Urban driving with conditional imitation learning." 2020 IEEE International Conference on Robotics and Automation (ICRA). IEEE, 2020.
>
> Imperfect Demonstrations:
>
> [4] Li, Ziniu, et al. "Imitation learning from imperfection: Theoretical justifications and algorithms." Advances in Neural Information Processing Systems 36 (2024).
>
> [5] Yang, Hanlin, Chao Yu, and Siji Chen. "Hybrid policy optimization from Imperfect demonstrations." Advances in Neural Information Processing Systems 36 (2024).

---

> ### Author Response · Authors · 2024-11-22
> **Thanks for your response!**
>
> Thank you for your valuable feedback and pointing out these relevant references. We have included these references and further strengthened the comparisons and discussions to related work and literature review in the revised paper.

---

### Official Review · Reviewer_eg39 · 2024-11-01

**Soundness:** 3
**Presentation:** 3
**Contribution:** 3
**Rating:** 6
**Confidence:** 3

**Summary:**

This paper introduces an IL method that handles imperfect demos by including both the optimal and non-optimal data in training, assuming that the demos come with a confidence score. The goal is to improve policy learning when demos aren’t perfectly optimal. The method assigns weights to optimal and non-optimal examples through a semi-supervised confidence classifier.  Experiments on six control tasks show that PN-GAIL performs better than standard GAIL and other baselines under these imperfect conditions.

**Strengths:**

* Originality: The use of positive and negative risks to manage imperfect demos tackles some of the real-world problems of non-optimal data in a practical way.
* Quality: The paper provides theoretical analysis for the positive-negative risk approach and shows experimental results across multiple control tasks.
* Clarity: Most theoretical ideas are clearly presented with good notation.
* Significance: The approach is relevant for real-world applications. It might have a real impact on the usability of IL in practical scenarios.

**Weaknesses:**

* The method’s reliance on confidence scores may limit its applicability when it’s difficult to assign confidence levels directly. Human annotation of preferences over trajectories, for instance, might be more feasible than assigning explicit confidence scores.
* The fundamental motivation of this paper is based on a claim that 'GAIL treats non-optimal demonstrations as if they were optimal'. I disagree with this claim. GAIL’s objective is to minimize the JS divergence between the expert and agent trajectory distributions, aiming to reproduce the overall distribution of expert demonstrations. When the agent policy is close to the expert policy, the discriminator's output tends to be 0.5 everywhere. If expert demos include a mix of optimal and sub-optimal trajectories, GAIL should naturally capture this mixture without necessarily assuming optimality. Could you provide a counterargument or clarification on why PN-GAIL’s approach is necessary, given this perspective?
* Lacks comparison with other advanced IL algorithms, such as f-IRL.

**Questions:**

* Theorem 1’s bound relies on knowing the variances. However, the variances can be difficult to estimate in real-world applications. Given this dependency, how practically useful is the bound in scenarios where variance values are unknown or hard to assess?
*  Theorem 2’s bound may become quite loose if the Rademacher complexity is high, as is typical with deep and wide neural networks. Could this have negative implications for the method's reliability when using complex models?
* Related to my disagree on the claim that 'GAIL treats non-optimal demonstrations as if they were optimal' in the weaknesses section, please could you provide a counterargument?

---

> ### Author Response · Authors · 2024-11-20
> **Author Response to Reviewer eg39**
>
> > **Q1.** The method's reliance on confidence scores may limit its applicability when it's difficult to assign confidence levels directly.
>
> **A1.** Thank you for the insightful comments. Our method needs to label a small percentage of confidence scores, which can be obtained by averaging multiple expert labels (as long as trajectory preferences are obtainable). To further validate the superiority of confidence scores over simple trajectory preference ranking, in **Common Response point 4**, we compare the performance of PN-GAIL with CAIL, TREX, and DREX (typical trajectory preference-based methods). The results show that PN-GAIL outperforms other methods, achieving the highest rewards.
>
>
>
> > **Q2.** Related to my disagree on the claim that 'GAIL treats non-optimal demonstrations as if they were optimal' in the weaknesses section, please could you provide a counterargument?
>
> **A2.** Thank you for this helpful comment. We have re-written this statement to avoid any potential confusion and also provided additional experimental results to further demonstrate this point. When expert demonstrations include non-optimal examples, GAIL aims to reproduce the overall distribution of these demonstrations, causing both optimal and non-optimal instances to be weighted equally. Consequently, if non-optimal demonstrations dominate, the learned policy will also reflect most non-optimal behaviors. In contrast, PN-GAIL mimics optimal demonstrations while minimizing the influence of non-optimal ones by assigning different weights to each type of demonstration and considering the associated positive and negative risks. This is also reflected in the poor performance of GAIL and the best performance of PN-GAIL in our additional experimental results in **Appendix B.3 Figure 8**.
>
>
> > **Q3.** Lacks comparison with other advanced IL algorithms, such as f-IRL.
>
> **A3.** Following your constructive comments, we have demonstrated the performance of f-IRL [1] in **Common Response point 5**. However, since f-IRL is not designed for imperfect demonstrations, it does not work well because it reproduces all demonstrations with the same weight.
>
>
> > **Q4.** Theorem 1's bound relies on knowing the variances, how practically useful is the bound in scenarios where variance values are unknown or hard to assess?
>
> **A4.** We are not sure if you are referring to Theorem 4.4 (If not, please let us know). For Theorem 4.4, the bound given is related to $\alpha$ and $\beta$. For computational convenience, we assume that these covariances are sufficiently small, and take a fixed approximation of $\alpha$ and $\beta$. The experiments in **Common Response point 6** verifies the rationality of the approximation. When $\alpha$ and $\beta$ use a fixed approximation, the bound given is independent of variance.
>
>
>
>
> > **Q5.** Theorem 2's bound may become quite loose if the Rademacher complexity is high.
>
> **A4.** Thank you for your insights. This phenomenon is indeed possible, and we believe that overfitting is one of the contributing factors. However, it can be mitigated through techniques such as regularization or dropout. Additionally, we can select different classification models based on the size of the dataset. For larger datasets, a more complex model may be appropriate, as the abundance of data helps reduce the risk of overfitting. Conversely, for smaller datasets, it is advisable to opt for a simpler model.
>
>
>
> [1] Tianwei Ni, Harshit Sikchi, Yufei Wang, Tejus Gupta, Lisa Lee, and Ben Eysenbach. f-irl: Inverse reinforcement learning via state marginal matching. In Conference on Robot Learning, pp. 529–551. PMLR, 2021.

---

> > ### Comment · Reviewer_eg39 · 2024-11-28
> >
> > Apologies for the delayed response. The author made valid points, so I am raising my score to 6. Good luck!

---

> > > ### Author Response · Authors · 2024-11-28
> > > **Thanks for your response!**
> > >
> > > We really appreciate your time for reviewing our paper and reading the follow-up rebuttal! If you had any additional questions, please feel free to ask any time.

---

### Official Review · Reviewer_H316 · 2024-11-05

**Soundness:** 3
**Presentation:** 4
**Contribution:** 4
**Rating:** 8
**Confidence:** 4

**Summary:**

This work addresses imitation learning from imperfect demonstrations, utilizing both confidence-labeled noisy demonstrations and unlabeled noisy demonstrations. It aims to overcome two primary limitations of prior work, specifically 2IWIL [A], a representative approach to this problem that employs a two-step learning process: (i) semi-confidence labeler training on the unlabeled dataset, and (ii) confidence-based generative imitation learning.

The proposed method, PN-GAIL (Positive-Negative GAIL), tackles the limitations of 2IWIL as follows:

1. **Incorporating Negative Risk to Objective**: 2IWIL overlooks the negative risk associated with imperfect demonstrations, leading the discriminator to disproportionately prioritize the positive risk of frequent samples. PN-GAIL addresses this by incorporating both positive and negative risks into the confidence-based imitation learning objective, ensuring a more reliable evaluation regarding to demonstration quality.
2. **Balanced Semi-Confidence (BSC) Classification**: In 2IWIL, semi-confidence (SC) classification is used to train a confidence labeler for unlabeled demonstrations. However, SC classification tends to overestimate the confidence of labeled data and underestimate the confidence of unlabeled data. To address this, PN-GAIL introduces a balanced semi-confidence (BSC) objective and further suggests near-optimal values for hyperparameters $\alpha$ and $\beta$, enhancing practical applicability.

[A] Wu et al., "Imitation Learning from Imperfect Demonstration," ICML 2019.

**Strengths:**

This work is well-motivated and effectively addresses the challenges presented in the previous study using sound methods.

A notable strength of this work is its practice-oriented design of the objective functions, which enhances applicability in real-world scenarios.
This study removes the dependence on $\eta$, the class prior $p(y=0)$ for imperfect demonstration datasets, from the primary objective.
Since $\eta$ is generally unknown and challenging to estimate, prior work treated it as a hyperparameter, requiring practitioners to invest substantial effort in tuning it.
By eliminating this reliance, the proposed approach reduces the overhead associated with hyperparameter optimization.

Additionally, the authors introduce near-optimal and practical choices for the parameters $\alpha$ and $\beta$ in the Balanced Semi-Confidence (BSC) objective, which can be straightforwardly calculated based on the dataset sizes $n_c$ (confidence-labeled) and $n_u$ (unlabeled).
This adjustment simplifies the implementation process and supports the broader applicability of imitation learning with imperfect demonstrations in practical settings.

Furthermore, the manuscript includes theoretical analyses showing that (i) the proposed objective helps avoid the imitation of non-optimal data and (ii) derives a sample complexity bound for the BSC method, providing a rigorous foundation for the proposed improvements.

**Weaknesses:**

Despite the many strengths of this work, the empirical results presented in the manuscript do not stand out as particularly impressive.

Specifically, Figure 1 shows that the performance difference between PN-GAIL and the most competitive baseline across tasks is not significant. In my opinion, as discussed in Section 2, since baseline methods typically assume a dominant proportion of $\pi_{opt}$, exploring scenarios with a more skewed ratio (e.g., $\pi_{opt}:\pi_1=1:10$) might provide a more notable results where conventional methods fail while PN-GAIL successes.
I think conducting experiments with more extreme demonstration ratios could more clearly demonstrate the scenarios in which PN-GAIL offers a distinct advantage over baseline methods.

**[Minor Comments]**
1. For Figures 2, 3, and 4, using distinct line colors for different methods would enhance readability.
2. In the ablation study presented in Figure 3, it would be advantageous to include results from 2IWIL—even though they are already provided in Figure 2. Since 2IWIL represents a variant of PN-GAIL that excludes both the PN objective and the BSC, its direct comparison within the same figure would clarify the incremental benefits of these components.

**Questions:**

Q1. In contrast to scenarios where $\pi_1$ dominates, how would the results in Figure 2 be affected if $\pi_{opt}$ were dominant, a condition under which existing methods are known to perform well? Results for PN-GAIL in the Pendulum-v1 task with various $\pi_{opt}:\pi_1$ ratios are presented in Figure 7 of the supplementary material, but a more systematic comparison between PN-GAIL and baseline methods could strengthen the manuscript. If PN-GAIL demonstrates robust performance and outperforms the baseline methods in such scenarios, it would confirm the method's reliability. This evidence would support the assertion that PN-GAIL performs consistently well across a range of dataset quality distributions, thus setting it apart from existing methods.

Q2. Instead of modifying the $\pi_{opt}:\pi_1$ ratio while maintaining a fixed total number of demonstrations, what would occur if the optimal demonstrations was kept invariant while the number of suboptimal demonstrations was varied? This setup would illustrate how PN-GAIL effectively utilizes additional suboptimal demonstrations, isolating the influence of optimal demonstrations on imitation performance. It would offer valuable insights into PN-GAIL’s ability to adapt to and leverage diverse demonstration qualities effectively.

**Details Of Ethics Concerns:**

I have no ethical concerns on this work.

---

> ### Author Response · Authors · 2024-11-20
> **Author Response to Reviewer H316**
>
> > **Q1.** Conducting experiments with more extreme demonstration ratios could more clearly demonstrate the scenarios in which PN-GAIL offers a distinct advantage over baseline methods.
>
> **A1.** In the Pendulum-v1 environment, the ratio of the demonstrations is $\pi_{\mathrm{opt}}:\pi_{1}=1:4$, in which case the advantage of PN-GAIL over other baseline methods is clearly visible. In addition, following your advice, we have added experiments at the ratio of $\pi_{\mathrm{opt}}:\pi_{1}=1:10$ in **Common Response point 1**. We notice that WGAIL performs even worse than GAIL, which we attribute to the fact that WGAIL incorrectly assigns high confidence scores to non-optimal demonstrations, resulting in a worse policy.
>
>
> > **Q2.** Modifications of Figures 2, 3, 4.
>
> **A2.** Thank you for your valuable comments. We have redrawn Figures 2, 3, 4 and also included the results of 2IWIL in Figure 3, following your advice. The redrawn plots have been placed in the revised version.
>
>
> > **Q3.** How would the results in Figure 2 be affected if $\pi_\mathrm{opt}$ were dominant?
>
> **A3.** In the Ant-v2 and Pendulum-v1 environments, we supplement the experiments at the demonstration ratio of $\pi_\mathrm{opt}:\pi_1=2:1$ in **Common Response point 2**.
>
>
> > **Q4.** What would occur if the optimal demonstrations were kept invariant while the number of suboptimal demonstrations varied?
>
> **A4.** We have conducted experiments with different numbers of non-optimal demonstrations in **Common Response point 3**. We find that when the number of non-optimal demonstrations decreases, the performance of PN-GAIL does not decrease significantly.

---

> > ### Comment · Reviewer_H316 · 2024-11-20
> >
> > Thank you for addressing my questions. I appreciate the inclusion of new experimental results, which now appear clearly convincing to align with the arguments presented in this research.
> > I have no further concerns and update my rating to 8, as the quality of the revision is strong enough to make a valuable contribution to this community.

---

> > > ### Author Response · Authors · 2024-11-20
> > > **Thanks for your response!**
> > >
> > > We greatly appreciate your time in reviewing our paper and reading the follow-up rebuttal! We're thrilled for your recognition of our work, thanks a lot!

---

### Author Response · Authors · 2024-11-20
**Common Response**

We sincerely appreciate the insightful feedback from the reviewers. **The revised paper based on the feedback has been uploaded.** We first respond to the common concerns and present the results of the experiments that we have supplemented.

**1.** Extreme optimal and non-optimal demonstration rate experiments.

To highlight the strength of PN-GAIL in an extreme optimal or non-optimal demonstration rate setting, in the Ant-v2 environment, we collect demonstrations with ratio of $\pi_\mathrm{opt}:\pi_1=1:10$. We then evaluate the performance of PN-GAIL and other baseline methods using the collected demonstrations. As shown in **Appendix B.3 Figure 8**, under this extreme demonstration ratio, all other methods fail, resulting in outcomes that are close to GAIL. In contrast, PN-GAIL successfully learns valuable information from the extreme demonstrations.

**2.** Experiments when optimal demonstrations are dominant.

To illustrate that PN-GAIL consistently performs well across a range of dataset quality distributions, in the Ant-v2 and Pendulum-v1 environments, we conduct the experiments at the demonstration ratio of $\pi_\mathrm{opt}:\pi_1=2:1$. As shown in **Appendix B.3 Figure 9**, it can be seen that when the optimal demonstrations are dominant, PN-GAIL still shows robust and excellent performance.

**3.** Experiments under different numbers of non-optimal demonstrations.

In order to understand how PN-GAIL can take advantage of additional non-optimal demonstrations, we conduct experiments with different numbers of non-optimal demonstrations in the Ant-v2 environment, and the results are presented in **Appendix B.3 Figure 10**. We find that when the number of non-optimal demonstrations decreases, the performance of PN-GAIL does not decrease significantly.

**4.** Comparative experiments with methods based on expert preference.

We compare the performance of PN-GAIL with CAIL, TREX, and DREX (typical expert preference-based methods) in the Ant-v2 and Pendulum-v1 environments. PN-GAIL outperforms the other methods, achieving the highest returns. More details can be seen in **Appendix B.3 Table 4**.

| Methods | Pendulum-v1 | Ant-v2 |
|:--------:| :---------:|:--------:|
| PN-GAIL | **-465.978±132.710** | **2960.458±851.465** |
| CAIL | -697.821±91.459 | 2336.546±735.378 |
| T-REX | -1074.435±258.078 | -1858.025±217.228 |
| D-REX | -1532.958±114.13 | -2495.473±220.703 |
| Optimal Policy | -116.81 | 4271.79 |

Table R1. Average returns of PN-GAIL, CAIL, T-REX and D-REX. Based on the implementation here: https://github.com/Stanford-ILIAD/Confidence-Aware-Imitation-Learning.

**5.** Comparative experiments with other advanced IL algorithms.

We compare the performance of PN-GAIL with f-IRL [1] in the Ant-v2 and Pendulum-v1 environments. As shown in the table below, PN-GAIL performs better than FKL(f-IRL), RKL(f-IRL) and JS(f-IRL). More details can be seen in **Appendix B.3 Table 4**.

| Methods | Pendulum-v1 | Ant-v2 |
|:--------:| :---------:|:--------:|
| PN-GAIL | **-465.978±132.710** | **2960.458±851.465** |
| FKL(f-IRL) | -698.064±300.793 | 1563.796±1372.482 |
| RKL(f-IRL) | -603.760±189.727 | 962.785±816.547 |
| JS(f-IRL) | -581.602±185.096 | 882.026±690.371 |
| Optimal Policy | -116.81 | 4271.79 |

Table R2. Average returns of PN-GAIL and f-IRL. Based on the implementation here: https://github.com/twni2016/f-IRL .

**6.** The variance of the estimator $\hat{R}_{BSC,\ell}(g)$ before and after inclusion $\alpha$ and $\beta$.

 We validate the appropriateness of our chosen values of $\alpha$ and $\beta$ by comparing the variance of the estimator $\hat{R}_{BSC,\ell}(g)$ before and after the inclusion of $\alpha$ and $\beta$.

| Var | Ant-v2 | HalfCheetah-v2 | Hopper-v2 | Pendulum-v1 | Swimmer-v2 | Walker2d-v2 |
|:--------:| :---------:|:--------:| :---------:| :---------:| :---------:| :---------:|
| Origin | 0.126±0.054 | 0.033±0.009 | 1.476±0.574 | 0.00064±0.00024 | 1.664±1.475 | 0.093±0.029 |
| PN-GAIL(Ours) | **0.100±0.050** | **0.025±0.006** | **1.412±0.566** | **0.00013±0.00004** | **0.007±0.015** | **0.084±0.028** |

Origin indicates the lack of $\alpha$ and $\beta$ (i.e., $\alpha$=0 and $\beta$=0). The results in the above table show that the variance of PN-GAIL is consistently smaller, demonstrating the validity of the chosen values for $\alpha$ and $\beta$.

**7.** Modification of the content of the article:
- We have redrawn the figures of the article to provide a clearer explanation and to enhance readability.
- We have added explanations for Figures 1 and 3 in Sections 4.1 and 5.1, respectively, for improving understanding.
- We have modified the undefined or unclearly defined parts such as the definitions of $\delta$ and PN-GAIL$\backslash$BSC.


[1] Tianwei Ni, Harshit Sikchi, Yufei Wang, Tejus Gupta, Lisa Lee, and Ben Eysenbach. f-irl: Inverse reinforcement learning via state marginal matching. In Conference on Robot Learning, pp. 529–551. PMLR, 2021.

---

### Meta-Review · Area_Chair_GPhv · 2024-12-20

**Metareview:**

This paper presents a well-motivated and theoretically sound approach to imitation learning with imperfect demonstrations, offering a valuable contribution to the field. The authors effectively address a key limitation of prior work by removing the reliance on the unknown class prior and introducing practical parameter choices.  This significantly improves the applicability and ease of use of the proposed method. The theoretical analyses provide further support for the approach. While the empirical results, in their current form, do not fully showcase the method's potential advantages compared to baselines, the concerns raised are relatively minor and could be addressed through additional experiments.  Given the strong theoretical grounding, practical improvements, and overall clarity of the paper, we recommend acceptance. The authors are encouraged to consider expanding the experimental evaluation to more definitively highlight the advantages of their method in scenarios where existing methods struggle.

**Additional Comments On Reviewer Discussion:**

nothing concerning

---

### Decision · Program_Chairs · 2025-01-22

Accept (Poster)